# TERRA R-loops trigger a switch in telomere maintenance towards break-induced replication and PRIMPOL-dependent repair

Suna In[1], Patricia Renck Nunes[1,2], Rita Valador Fernandes[1,3] & Joachim Lingner [1✉]

## Abstract

TERRA long noncoding RNAs associate with telomeres post transcription through base-pairing with telomeric DNA forming R-loop structures. TERRA regulates telomere maintenance but its exact modes of action remain unknown. Here, we induce TERRA transcription and R-loop formation in telomerase-expressing cells and determine that TERRA R-loop formation requires non-redundant functions of the RAD51 DNA recombinase and its enhancer RAD51AP1. TERRA R-loops interfere with semiconservative DNA replication, promoting telomere maintenance by a homology-directed repair (HDR) mechanism known as break-induced replication (BIR), which ensures telomere maintenance in ALT cancer cells. In addition, TERRA induces PRIMPOL-dependent repair, which can initiate DNA synthesis de novo downstream of replication obstacles. PRIMPOL acts in parallel to BIR for telomere maintenance of TERRA-overexpressing cells, promoting their survival. Similarly, we find that PRIMPOL depletion is synthetic-lethal with BIR deficiency in U2OS ALT cancer cells. Therefore, TERRA R-loops by themselves are sufficient to induce ALT-typical telomere repair mechanisms, in the absence of other ALT-typical telomeric chromatin changes.

**Keywords** Break-induced Replication; PRIMPOL; R-loops; Telomeres; TERRA
**Subject Categories** Chromatin, Transcription & Genomics; DNA Replication, Recombination & Repair

## Introduction

Telomeric repeat containing RNAs (TERRA) are long noncoding RNAs that are transcribed from subtelomeric promoters towards chromosome ends (Azzalin et al, 2007; Feretzaki et al, 2019; Nergadze et al, 2009). TERRA molecules consist of chromosome end-specific subtelomeric sequences followed by telomeric UUAGGG repeats (Porro et al, 2014). TERRA can associate with chromosome ends through formation of RNA:DNA hybrid structures giving rise to R-loops, in which the telomeric G-rich DNA strand is displaced (Feretzaki et al, 2020). Intriguingly, TERRA R-loops can form post transcription in dependency of the RAD51 DNA recombinase (Feretzaki et al, 2020). RAD51 is bound to TERRA in nuclear extracts and it can catalyze R-loop formation in vitro suggesting that a RAD51-mediated homology search mechanism enables TERRA association with chromosome ends post transcription.

In telomerase-positive cancer cells and possibly most healthy human cells, TERRA is repressed in S phase, probably to prevent the interference of TERRA R-loops with semiconservative DNA replication (Porro et al, 2010). However, the repression of TERRA in S phase is lost in alternative lengthening of telomeres (ALT) cancer cells (Flynn et al, 2015) promoting the use of a telomerase-independent mechanism to counteract the end replication problem (Arora et al, 2014). Thus, TERRA provides oncogenic functions in ALT cancer (Kyriacou and Lingner, 2024). Increased TERRA expression and telomeric R-loops during S phase also occur at short telomeres in *Saccharomyces cerevisiae* cells from which telomerase was deleted (Graf et al, 2017). Significantly, activation of HDR at short telomeres required TERRA R-loops, which persisted in S phase slowing down telomere shortening. Similarly, TERRA R-loops were persistent in human cells that approached cellular senescence (Sze et al, 2023). In these cells, TERRA-mediated sister telomere cohesion appeared to contribute either to telomere maintenance or telomere capping. Thus, in these examples, TERRA R-loops counteract short telomeres-induced cellular aging.

ALT is mostly seen in cancers of mesenchymal origin (Pickett and Reddel, 2015). ALT utilizes homology-directed repair (HDR) pathways to lengthen telomeres. Upon strand invasion of one telomere into another, unidirectional conservative telomeric DNA synthesis occurs through the engagement of a PCNA-DNA polymerase δ (Polδ) containing replisome (Dilley et al, 2016). This HDR pathway resembles break-induced replication (BIR), which has been thoroughly characterized in *S. cerevisiae* (Epum and Haber, 2022). However, HDR is repressed at intact healthy telomeres in normal cells and telomerase-expressing cancer cells (Sfeir et al, 2010; Glousker et al, 2020). To overcome HDR repression, ALT telomeres undergo extensive changes in telomeric DNA structure and protein composition. DNA damage response

[1]Swiss Institute for Experimental Cancer Research (ISREC), School of Life Sciences, École Polytechnique Fédérale de Lausanne (EPFL), Lausanne 1015, Switzerland. [2]Present address: Bristol Myers Squibb, Lawrence Township, NJ 08648, USA. [3]Present address: IQVIA, Lagoas Park, Porto, Salvo 2740-266, Portugal. ✉E-mail: joachim.lingner@epfl.ch

proteins and mediators of HDR appear, and the histone variant H3.3 disappears at ALT telomeres (Déjardin and Kingston, 2009; Zhang et al, 2023; Kaminski et al, 2022). ALT telomeres also associate with PML proteins in subnuclear bodies referred to as ALT-associated PML bodies that mediate telomeric DNA synthesis. Elevated TERRA expression and increased TERRA R-loops are another feature of ALT cells and were proposed to promote recombination at ALT telomeres. Indeed, regulated levels of telomeric RNA-DNA hybrids at ALT telomeres are crucial to exert the ALT-typical DNA replication stress at telomeres and stimulate HDR (Arora et al, 2014; Silva et al, 2021, 2022, 2019). However, the extent to which the observed effects of TERRA are influenced by the unique composition of telomeric chromatin in ALT cells remained unclear.

In this study, we elucidate the roles of TERRA in telomere regulation by investigating the effects of increased TERRA transcription in non-ALT telomerase-expressing cancer cells (HeLa). Using CRISPR activation to induce TERRA expression, we examine its impact on telomeric R-loop formation, DNA damage, and telomere fragility. We find that RAD51 and RAD51AP1 provide non-redundant roles to mediate TERRA R-loop formation. TERRA R-loops induce telomere damage, which is repaired either by break-induced replication or PRIMPOL-dependent repriming resolving replication stress and maintaining telomere integrity. We also find that PRIMPOL contributes to the survival of U2OS ALT cancer cells. Thus, TERRA-induced telomere damage triggers DNA repair pathways that are required for the cell survival of ALT cancer cells.

# Results

## TERRA overexpression leads to an increase in telomeric R-loops

To investigate the roles of TERRA in telomere regulation, we engineered HeLa cells for TERRA overexpression using CRISPR activation (Fig. 1A). To this end, we expressed four copies of VP16 (VP64), a strong transcription activator, in frame with catalytically dead Cas9 (dCas9) from a lentiviral vector. We also stably expressed either two guide RNAs, which were designed to target TERRA promoter regions at multiple chromosome ends (Fig. 1A, "Reagents and Tools Table"), or a control guide RNA targeting the AAVS site (control cells) ("Reagents and Tools Table"). HeLa cells expressing TERRA promoter-targeting guide RNAs exhibited a ~40-fold increase in TERRA expression compared to cells expressing the control guide RNA (Fig. 1B). This expression level is similar to the range seen in ALT cells (U2OS, VA13, GM847, SAOS-2) that were analyzed in parallel and which displayed ~25–48-fold higher TERRA levels than HeLa control cells (Fig. 1B). CRISPR activation-induced TERRA overexpression was confirmed through RT-qPCR amplifying various subtelomeric TERRA sequences, demonstrating locus-specific TERRA induction (Fig. 1C). As expected, only subtelomeres with guide RNA binding sites showed an increase in TERRA levels (Fig. 1A,C). Consistent with VP16-mediated TERRA promoter activation, Chromatin Immunoprecipitation (ChIP) using an anti-acetyl-H4 antibody revealed increased H4 acetylation at guide RNA-targeted subtelomeres (Fig. EV1A) but not within TTAGGG telomeric repeats (Fig. EV1B).

Next, we performed DNA:RNA hybrid Immunoprecipitation (DRIP) using the S9.6 antibody (Aguilera and Ruzov, 2022) to examine telomeric R-loops upon TERRA induction. As a control for the specificity of the assay, we treated prior to immunoprecipitation the isolated nucleic acids in vitro with RNaseH1, which is an enzyme that degrades the RNA moiety of R-loops. RNaseH1 treatment abolished the DRIP signal, confirming that the immunoprecipitates contained R-loops (Fig. 1D). As expected, cells overexpressing TERRA showed higher levels of TERRA R-loops at telomeres compared to control cells (Fig. 1D,E). Notably, the 7p subtelomere, which does not overexpress TERRA, also formed more R-loops upon TERRA induction, supporting the notion that TERRA R-loops form post transcription in trans (Fig. 1E) (Feretzaki et al, 2020). Overall, the data demonstrate that the TERRA induction system in HeLa cells significantly increases endogenous TERRA expression and telomeric R-loops.

## RAD51 and RAD51AP1 are required for R-loop formation

Our recent study demonstrated that the DNA recombinase RAD51 promotes telomeric R-loop formation in HeLa cells (Feretzaki et al, 2020). However, in other works, the RAD51-interacting protein RAD51AP1 has been reported to promote TERRA R-loops in vitro (Yadav et al, 2022) and in ALT cells in order to facilitate the alternative lengthening of telomeres through its interaction with TERRA (Kaminski et al, 2022). We investigated whether RAD51AP1 plays a similar role in telomerase-positive cells. To this end, we depleted RAD51 and RAD51AP1 in HeLa cells (Fig. 2A) and performed DRIP analyses (Fig. 2B). Knockdown of either RAD51 or RAD51AP1 significantly suppressed telomeric R-loop formation in TERRA-overexpressing cells (Fig. 2B). This indicates that both RAD51 and RAD51AP1 are crucial for the formation of telomeric R-loops under conditions of high TERRA expression. Furthermore, this data provides novel evidence for the role of RAD51AP1 in non-ALT HeLa cells and demonstrates that RAD51 and RAD51AP1 play non-redundant roles in TERRA R-loop formation. Notably, while the knockdown efficiency was similar in control cells, the depletion did not affect TERRA R-loop levels. It is possible that the majority of basal R-loops in the here-used HeLa cells with a relatively long telomere length of ~10 kb are formed co-transcriptionally, involving the invasion of nascent RNA into DNA in a RAD51-independent manner. Of note, HeLa cells with short telomeres contain more R-loops and show RAD51-dependent TERRA R-loop formation (Feretzaki et al, 2020). Taken together, our findings demonstrate that both RAD51 and RAD51AP1 are essential for telomeric R-loop formation in HeLa cells with elevated TERRA levels.

## TERRA induces DNA damage at telomeres

R-loops are thought to potentially interfere with the replication machinery, and they may contribute to DNA damage (García-Muse and Aguilera, 2019). To investigate this for TERRA R-loops, we assessed the frequency of telomere dysfunction-induced foci (TIFs) by performing immunofluorescence (IF) using an antibody against 53BP1, a marker of DNA damage, followed by telomeric fluorescent in situ hybridization (FISH) (Fig. 3A,B). We evaluated the number of cells with more than five 53BP1 foci colocalizing with telomeric foci (Takai et al, 2003). Upon TERRA induction, there was a

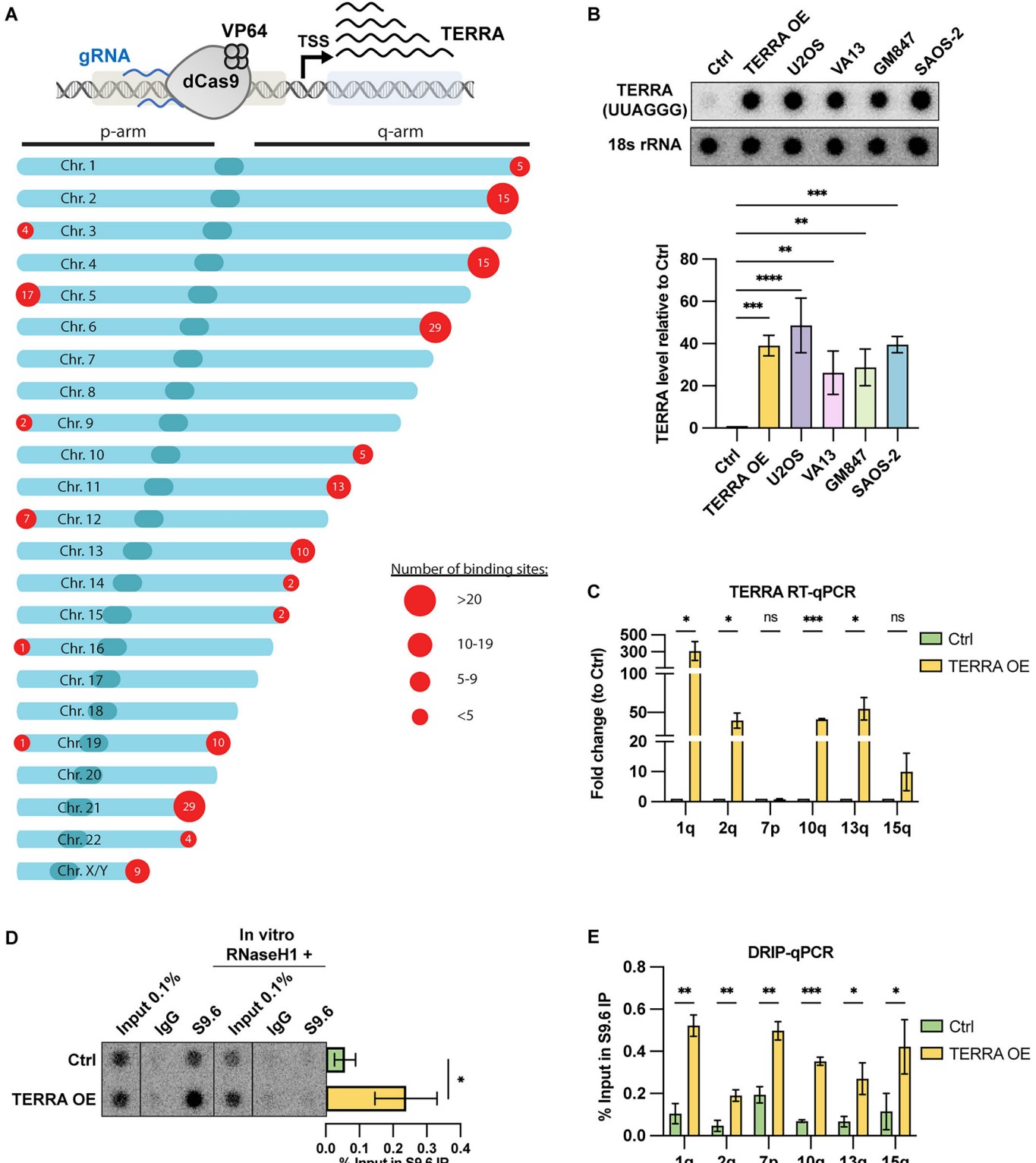

marked increase in the fraction of cells exhibiting more than five TIFs, comparable to the increase observed with zeocin treatment, which induces telomere damage as well as DNA double-stranded breaks throughout the genome. Thus, TERRA overexpression alone induces significant DNA damage at telomeres (Fig. 3A,B). In

addition, cells with more than five non-telomeric 53BP1 foci were significantly increased by zeocin treatment, but not by TERRA overexpression (Fig. EV2A,B). Thus, TERRA induces DNA damage specifically at telomeres. To determine whether TERRA-induced telomeric DNA damage results from R-loops, we ectopically

**Figure 1. TERRA overexpression leads to an increase in telomeric R-loops.**

(A) Schematic representation of dCas9-VP64-induced expression of endogenous TERRA (top) and number of potential guide RNA binding sites near TERRA promoters (bottom). (B) RNA dot blot analysis (upper panel) of control and TERRA-overexpressing HeLa cells (containing average telomere length 10 kb) and various ALT cell lines (U2OS, VA13, GM847, and SAOS-2). Quantification of signal intensity (lower) is presented as fold change relative to control cells and normalized to 18S rRNA levels. Data represent the mean ± s.d. from three independent biological replicates. One-way analysis of variance (ANOVA) with Dunnett's multiple comparisons test was applied. *P* values from left to right: \*\*\**P* = 0.0004, \*\*\*\**P* < 0.0001, \*\**P* = 0.0093, \*\**P* = 0.0047, \*\*\**P* = 0.0003. (C) Quantification of TERRA levels by RT-qPCR analyzed using indicated subtelomeric primers, plotted as fold change over Ctrl and normalized to GAPDH RNA levels. Data represent mean ± s.d. from three independent biological replicates. Multiple unpaired *t* test was applied. *P* values from left to right: \**P* = 0.009, \**P* = 0.003, ns *P* = 0.153, \*\*\**P* < 0.001, \**P* = 0.003, ns *P* = 0.067. (D, E) DRIP assay using S9.6 antibody. DRIP samples and inputs were treated with RNase (DNase-free) and analyzed by DNA dot blot with a ³²P-radiolabeled telomeric probe (D) or by qPCR with indicated subtelomeric primers (E). As a negative control, samples treated in vitro with RNaseH1 prior to immunoprecipitation were analyzed in parallel. (D) Data represent mean ± s.d. from three independent biological replicates. Unpaired *t* test was applied. \**P* = 0.0321 (E) Data represent mean ± s.d. from three independent biological replicates. Multiple unpaired *t* test was applied. *P* values from left to right: \*\**P* = 0.0005, \*\**P* = 0.0028, \*\**P* = 0.0008, \*\*\**P* < 0.0001, \**P* = 0.0112, \**P* = 0.0265. Source data are available online for this figure.

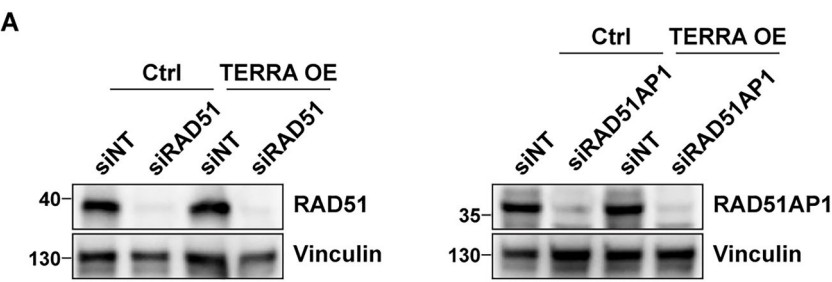

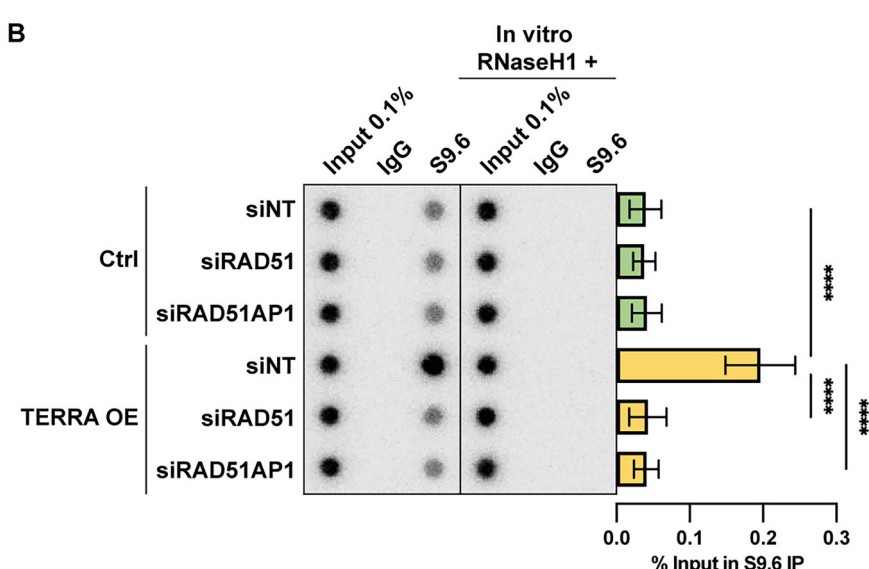

**Figure 2. RAD51 and RAD51AP1 are required for TERRA R-loop formation.**

(A) Western blot analysis upon depletion of RAD51 (left) or RAD51AP1 (right) in control and TERRA-overexpressing HeLa cells. (B) DRIP assay using S9.6 antibody. DRIP samples and inputs were treated with RNase (DNase-free) and analyzed by DNA dot blot with a ³²P-radiolabeled telomeric probe. As a negative control, samples treated in vitro with RNaseH1 prior to immunoprecipitation were analyzed in parallel. Data represent mean ± s.d. from five independent biological replicates. Two-way analysis of variance (ANOVA) with Tukey's multiple comparisons test was applied. *P* values from upper to lower: \*\*\*\**P* < 0.0001, \*\*\*\**P* < 0.0001, \*\*\*\**P* < 0.0001. Source data are available online for this figure.

expressed RNaseH1 (Fig. EV2C) and assessed telomere TIFs. Notably, RNaseH1 overexpression completely abolished the increase in TIFs observed upon TERRA induction, while no effect was seen in control cells (Fig. 3C). These findings strongly suggest that TERRA R-loops are the primary drivers of

the increased DNA damage at telomeres seen in TERRA-overexpressing cells.

To further characterize the TERRA R-loop-induced DNA damage, we inhibited the DNA checkpoint kinases ATM and ATR with small molecule inhibitors (Fig. 3D). ATM is activated by

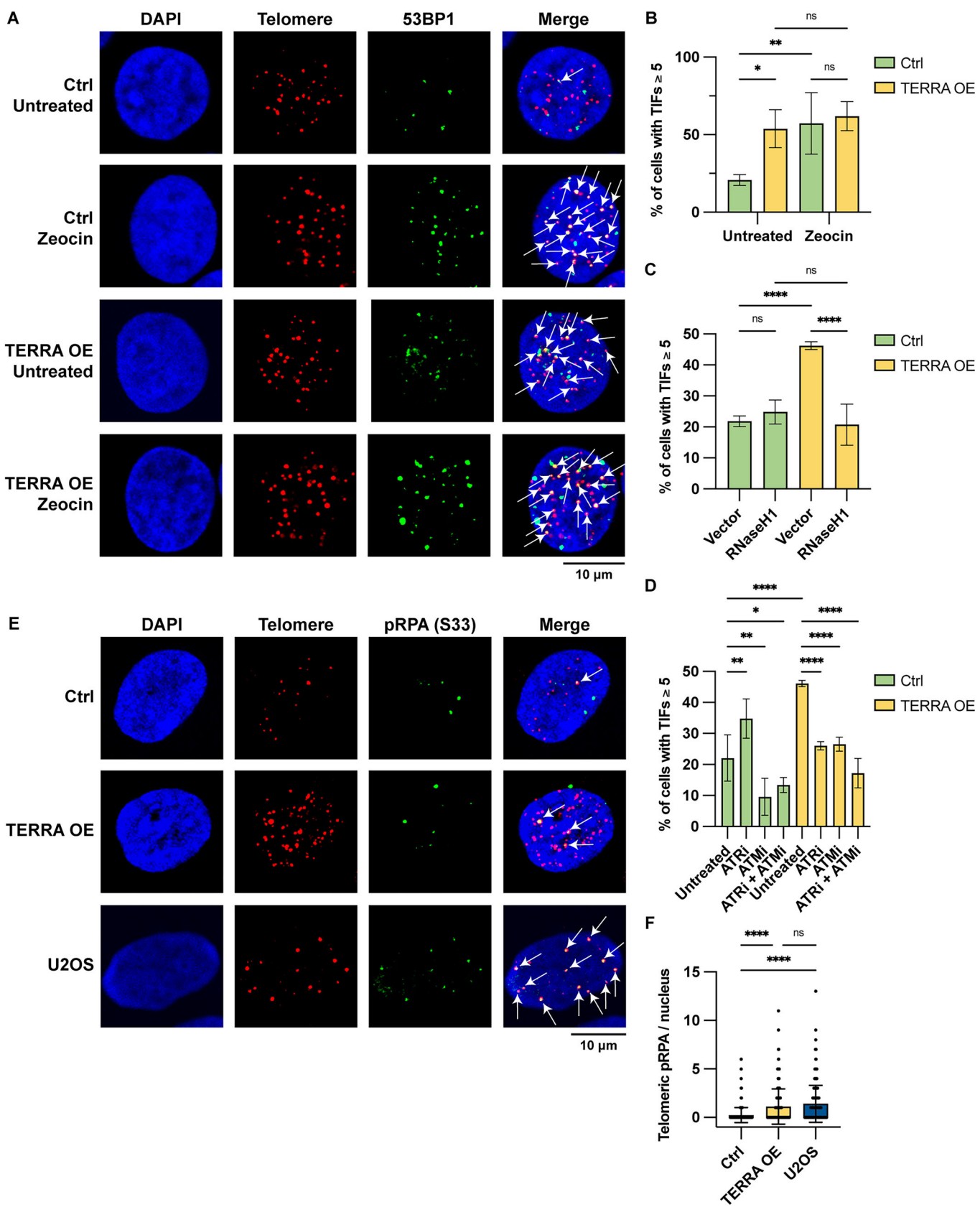

Figure 3. TERRA induces telomere dysfunction-induced foci (TIFs).

(A) Representative images for detection of 53BP1 at telomeres in control and TERRA-overexpressing HeLa cells. Immunofluorescence (IF) for 53BP1 (green) was combined with telomeric (CCCTAA)$_3$-FISH (red) and DAPI staining (blue). White arrows indicate 53BP1 foci colocalizing with telomeres. (B) Quantification of the number of cells with ≥5 telomeres colocalizing with 53BP1. Data represent mean ± s.d. from three independent biological replicates. Two-way analysis of variance (ANOVA) with uncorrected Fisher's least significant difference (LSD) test was applied. P values from left to right: *P = 0.0126, **P = 0.0077, ns P = 0.6679, ns P = 0.4572. (C) Quantification of the number of cells with ≥5 telomeres colocalizing with 53BP1 upon RNaseH1 overexpression. Data represent mean ± s.d. from three independent biological replicates. Two-way analysis of variance (ANOVA) with uncorrected Fisher's least significant difference (LSD) test was applied. P values from left to right: ns P = 0.3822, ****P < 0.0001, ns P = 0.2428, ****P < 0.0001. (D) Quantification of the number of cells with ≥5 telomeres colocalizing with 53BP1 in the absence and presence of ATM and ATR inhibitors (KU-55933 and VE-821, respectively; 10 μM, 24-h treatment). Data represent mean ± s.d. from three independent biological replicates. Two-way analysis of variance (ANOVA) with uncorrected Fisher's least significant difference (LSD) test was applied. P values from left to right: **P = 0.0041, **P = 0.0036, *P = 0.0329, ****P < 0.0001, ****P < 0.0001, ****P < 0.0001, ****P < 0.0001. (E) Representative images for detection of phosphorylated RPA32 (Ser33) at telomeres in control and TERRA-overexpressing HeLa cells. Immunofluorescence (IF) for pRPA (green) was combined with telomeric (CCCTAA)$_3$-FISH (red) and DAPI staining (blue). White arrows indicate pRPA (S33) foci colocalizing with telomeres. (F) Quantifications of the number of pRPA foci colocalizing with telomeres per nucleus. Data are presented as a scatter plot with bars indicating the mean ± s.d. from three independent biological replicates. One-way analysis of variance (ANOVA) with Tukey's multiple comparisons test was applied. P values from left to right: ****P < 0.0001, ****P < 0.0001, ns P = 0.0578. Source data are available online for this figure.

DNA double strand breaks whereas ATR by elongated stretches of single-stranded DNA that may stem from replication stress or the nucleolytic processing of DNA double strand breaks. Inhibition of either ATM or ATR reduced TIFs in TERRA-overexpressing cells, indicating that TERRA R-loops promote accumulation of telomeric DNA double strand breaks as well as extended stretches of single-stranded telomeric DNA, both of which were borne out in the subsequent analyses.

Given the role of ATR in TIFs upon TERRA induction, we further investigated whether TERRA R-loop–induced DNA damage promotes the formation of phosphorylated RPA (pRPA) at telomeres, which is a hallmark of the DNA damage response to single-stranded DNA. Thus, we performed IF-FISH using a pRPA (Ser33) antibody (Fig. 3E,F), including U2OS cells as a positive control, which exhibited significant colocalization of pRPA with telomeres as previously reported (Silva et al, 2021). Notably, TERRA overexpression led to a marked increase in pRPA foci at telomeres (Fig. 3F), indicating elevated replication stress and ssDNA accumulation. These findings support a model in which TERRA R-loops induce telomeric DNA damage by generating replication-associated single-stranded DNA, thereby activating the DNA damage response.

## TERRA R-loops directly induce telomere fragility

R-loops formed during the S phase may potentially obstruct the replication machinery and contribute to DNA damage (García-Muse and Aguilera, 2019). Increased replication stress, induced by low doses of aphidicolin, which inhibits replicative DNA polymerases α and δ, has been reported to display telomere fragility (Garcia-Exposito et al, 2016; Glousker and Lingner, 2021), characterized by smeared or doubled telomeric FISH signals on metaphase chromosomes (Fig. 4A). Several studies have reported an increase in telomere fragility when TERRA R-loop levels are elevated (Feretzaki et al, 2020; Fernandes and Lingner, 2023; Petti et al, 2019; Silva et al, 2019; Lin et al, 2021; Arora et al, 2014). However, these studies often indirectly affected TERRA R-loops by perturbing TERRA R-loop-regulating factors or they were done in ALT cells with strongly altered telomeric structures and protein composition. A study that most directly assessed TERRA R-loops on telomere fragility in HeLa cells utilized transgenic TERRA expressed from extrachromosomal plasmids (Feretzaki et al, 2020). Yet, to rule out the minor possibility that the PP7-coat protein fused to GFP (PCP-GFP) tag in transgenic TERRA influences the

observed effects, it is crucial to evaluate telomere fragility driven specifically by endogenous TERRA R-loops.

We analyzed telomere fragility upon overexpression of endogenous TERRA using the above introduced CRISPR activation system and included TRF1 depletion (Fig. EV3A), which is known to induce telomere fragility (Sfeir et al, 2009), as a positive control (Fig. 4B). The basal level of telomere fragility was observed to range between 3 and 5% (Fig. 4B). Similar to TRF1 depletion, TERRA overexpression led to an approximately threefold increase in telomere fragility to a frequency of 12% (Fig. 4B). The combination of TERRA overexpression and TRF1 depletion showed a partially additive effect (Fig. 4B). Thus, induction of endogenous TERRA in HeLa cells increases telomere fragility and TERRA exacerbates the telomere instability in TRF1 compromised cells.

To determine if the observed fragility was due to R-loops, we transiently overexpressed RNaseH1 (Fig. EV3B) to induce degradation of TERRA in R-loops (Fig. 4C). RNaseH1 overexpression suppressed the telomeric R-loops while a catalytically dead RNaseH1 (D145N) did not affect TERRA R-loops (Fig. 4C). At the same time, RNaseH1 overexpression completely suppressed the increased telomere fragility obtained upon TERRA overexpression (Fig. 4D). Of note, although DRIP analysis showed a further reduction of R-loops below basal levels upon RNaseH1 over-expression in control cells (Fig. 4C), telomere fragility did not drop below the basal level of roughly 3–5% (Fig. 4D). This suggests that the basal level of telomere fragility in HeLa cells stems mainly from other replication stress sources (G-quadruplex structures, unresolved t-loops, oxidative and other DNA damage, etc.). Indeed, the low endogenous TERRA levels in HeLa cells are further suppressed in S phase (Porro et al, 2010). As RNaseH1 depletion (Fig. EV3C) in non-overexpressing cells increased telomere fragility to levels similar to TERRA overexpression (Fig. EV3D), with no additional effect when TERRA was also induced (Fig. EV3D), it seems that RNaseH1 suppresses R-loops in S-phase cells. Overall, our experiments demonstrate that the increased telomere fragility seen in TERRA-overexpressing cells is due to increased R-loop formation at telomeres and that RNaseH1 counteracts R-loops.

We also tested if RAD51 and RAD51AP1, both of which are required for TERRA R-loop formation, are required for telomere fragility. Strikingly, RAD51 or RAD51AP1 depletion suppressed the increase in fragility observed upon TERRA induction (Fig. 4E). In control cells, however, RAD51 and RAD51AP1 depletion caused a slight increase in fragility (Fig. 4E). This implies that RAD51 and

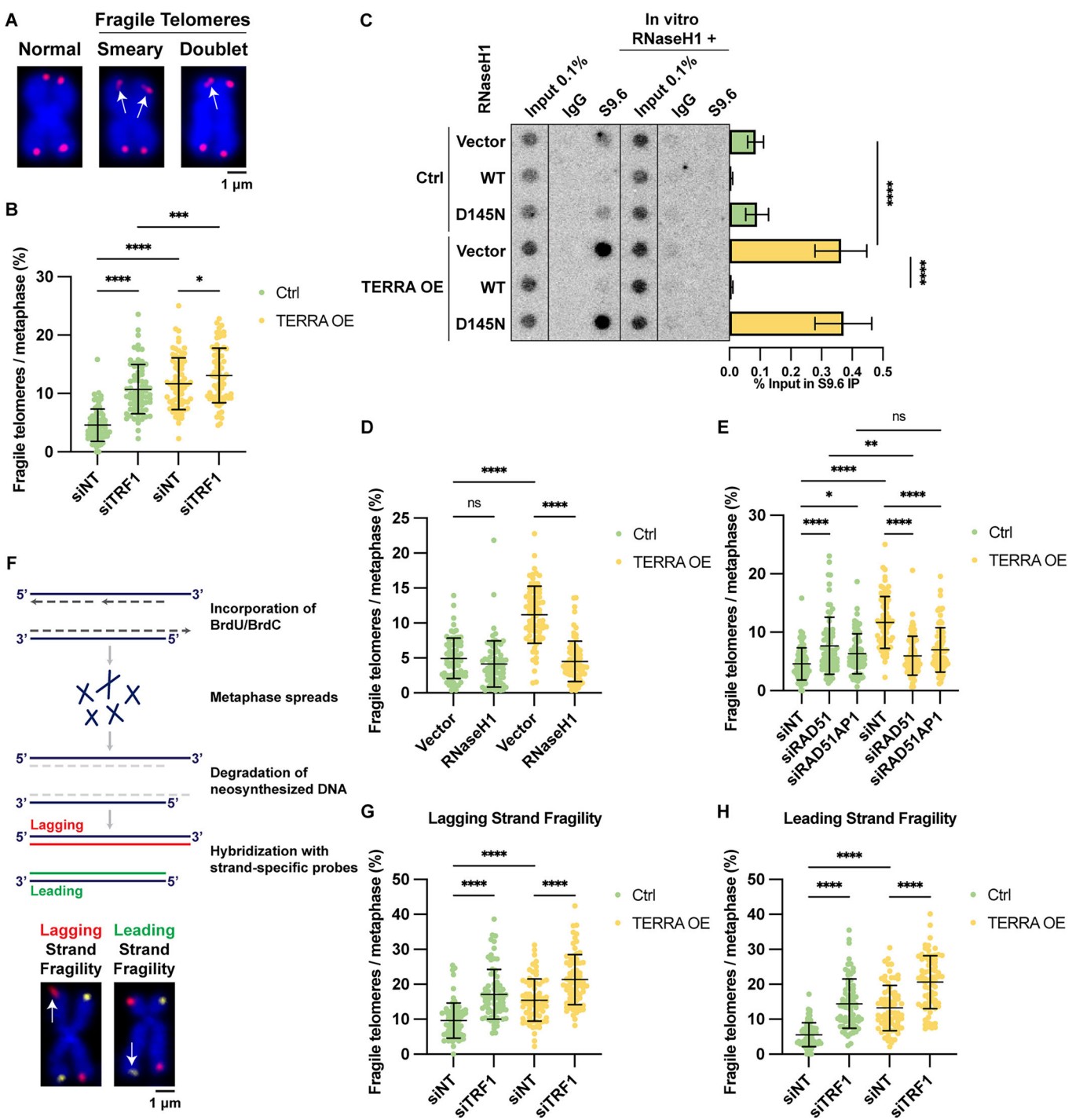

RAD51AP1 may undertake several roles in regulating telomere fragility and maintenance; when TERRA levels are high, RAD51 and RAD51AP1 facilitate post-transcriptional TERRA R-loop formation, which then becomes the main source of telomere fragility. Conversely, when TERRA levels are low, depletion of these factors may perturb telomere replication and repair pathways that overcome the already above-mentioned R-loop-independent replication obstacles (G-quadruplex structures, unresolved t-loops, oxidative and other DNA damage, etc.). Overall, our results demonstrate that TERRA-induced telomere fragility requires RAD51 and RAD51AP1-dependent R-loops.

## TERRA overexpression increases telomere fragility at both leading and lagging strand telomeres

During telomere replication, the parental telomeric C-rich strand serves as a template for leading strand synthesis while the G-strand serves as a template for lagging strand synthesis. TERRA will base-

**Figure 4. TERRA R-loops directly induce telomere fragility at both lagging and leading strands.**

(A) Representative images of metaphase spreads stained with telomeric (CCCTAA)$_3$-FISH probe (red) and DAPI (blue). While normal telomeres show a single round signal at the end of each chromosome arm (left), fragile telomeres have smeary (middle) or multiple telomeric signals (right). White arrowheads indicate fragile telomeres. (B) Quantification of telomere fragility upon depletion of TRF1 in control and TERRA-overexpressing HeLa cells. At least 25 metaphases were analyzed per condition per replicate, and three independent biological replicates were performed. Horizontal lines and error bars represent mean ± s.d. Two-way analysis of variance (ANOVA) with Tukey's multiple comparisons test was applied. P values from left to right: ****$P < 0.0001$, ****$P < 0.0001$, ***$P = 0.0005$, *$P = 0.0359$. (C) DRIP (DNA:RNA hybrid immunoprecipitation) assay using S9.6 antibody. DRIP samples and inputs were treated with RNase (DNase-free) and analyzed by DNA dot blot with a $^{32}$P-radiolabeled telomeric probe. As a negative control, samples treated in vitro with RNaseH1 prior to immunoprecipitation were analyzed in parallel. Data represent mean ± s.d. from four independent biological replicates. Two-way analysis of variance (ANOVA) with Tukey's multiple comparisons test was applied: P values from upper to lower: ****$P < 0.0001$, ****$P < 0.0001$. (D) Quantification of telomere fragility upon overexpression of RNaseH1 in control and TERRA-overexpressing HeLa cells. At least 25 metaphases were analyzed per condition per replicate, and three independent biological replicates were performed. Horizontal lines and error bars represent mean ± s.d. Two-way analysis of variance (ANOVA) with Tukey's multiple comparisons test was applied. P values from left to right: ns $P = 0.4679$, ****$P < 0.0001$, ****$P < 0.0001$. (E) Quantification of telomere fragility upon depletion of RAD51 or RAD51AP1 in control and TERRA-overexpressing HeLa cells. At least 25 metaphases were analyzed per condition per replicate, and three independent biological replicates were performed. Horizontal lines and error bars represent mean ± s.d. Two-way analysis of variance (ANOVA) with Tukey's multiple comparisons test was applied. P values from left to right: ****$P < 0.0001$, *$P = 0.0155$, ****$P < 0.0001$, **$P = 0.0067$, ns $P = 0.2408$, ****$P < 0.0001$, ****$P < 0.0001$. (F) (Top) Schematic of the CO-FISH experiment. HeLa cells were incubated with BrdU and BrdC for 15 h to ensure incorporation during a single round of replication. After metaphase enrichment via demecolcine treatment, cells were harvested, and metaphases were spread onto microscopic slides. Following RNaseA treatment and Hoechst staining, the newly synthesized DNA strands that incorporated BrdU and BrdC were degraded upon UV irradiation followed by Exonuclease III digestion. The remaining parental DNA strands were then hybridized with strand-specific probes. The blue lines indicate parental strands, the dashed gray lines indicate BrdU/BrdC incorporated newly synthesized DNA strands, the red line indicates lagging strand-specific probes, and the green line indicates leading strand-specific probes. (Bottom) Representative images of metaphase spreads stained with TYE563-TeloC LNA probe (red), FAM-TeloG LNA probe (yellow), and DAPI (blue). White arrowheads indicate fragile telomeres. (G) Quantification of telomere fragility at lagging strand upon depletion of TRF1 in control and TERRA-overexpressing HeLa cells with 30 kb telomeres. At least 25 metaphases were analyzed per condition per replicate, and three independent biological replicates were performed. Horizontal lines and error bars represent mean ± s.d. Two-way analysis of variance (ANOVA) with Tukey's multiple comparisons test was applied. P values from left to right: ****$P < 0.0001$, ****$P < 0.0001$, ****$P < 0.0001$. (H) Quantification of telomere fragility at the leading strand upon depletion of TRF1 in control and TERRA-overexpressing HeLa cells with 30 kb telomeres. At least 25 metaphases were analyzed per condition per replicate, and three independent biological replicates were performed. Horizontal lines and error bars represent mean ± s.d. Two-way analysis of variance (ANOVA) with Tukey's multiple comparisons test was applied. P values from left to right: ****$P < 0.0001$, ****$P < 0.0001$, ****$P < 0.0001$. Data from Figs. 4B,E and 5A,B,D stem from common experiments explaining the identity of the siNT control in these panels. Source data are available online for this figure.

pair with the C-rich strand when engaged in an R-loop structure which may interfere with leading strand synthesis. In addition, we speculated that the TERRA R-loop-displaced single-stranded G-rich telomeric DNA strand may form as collateral damage thermodynamically favored G-quadruplex structures (Yadav et al, 2022) which may interfere with lagging strand synthesis. Thus, in principle, the replication of both strands may be challenged by TERRA R-loops. Consistent with this notion, a recent study demonstrated that the accumulation of R-loops at telomeres induced by depletion of THOC subunits results in an increase of fragility of leading and lagging strand telomeres (Fernandes and Lingner, 2023). To assess the strand-specificity of telomere fragility upon TERRA induction, we performed telomeric chromosome orientation FISH (CO-FISH) (Bailey et al, 1996). In brief, HeLa cells were incubated with BrdU and BrdC for 15 h for incorporation of the modified nucleotides into the newly synthesized daughter strands (Fig. 4F). After spreading metaphase chromosomes on slides and staining with Hoechst dye, slides were UV irradiated, which preferentially damages the newly synthesized DNA strands containing the modified nucleotides. Following exonuclease III treatment, which degraded the nascent gap-containing DNA, the remaining G-rich parental strands were hybridized with 6-FAM-labeled LNA probes, while the parental C-rich strands were detected by hybridization with TYE563-labeled LNA probes (Fig. 4F). For optimal staining conditions, HeLa cells with >30 kb long telomeres were used to perform CO-FISH. Again, TRF1-depleted cells were used as a positive control. The basal level of telomere fragility detected by FISH was with ~9% higher in the 30 kb telomere cells (Fig. EV3E) than in cells with 10 kb telomeres (Fig. 4B), as already reported (Fernandes and Lingner, 2023). Nonetheless, HeLa cells with 30 kb telomeres also showed increased

telomere fragility upon TERRA induction or TRF1 depletion as seen in cells with 10 kb telomeres (Fig. EV3E). Furthermore, CO-FISH experiments demonstrated that TERRA-induced telomere fragility occurred at both leading and lagging strand telomeres (Fig. 4G,H), aligning with findings from THOC subunit depletion studies (Fernandes and Lingner, 2023) as well as fragility obtained by TRF1 depletion (Fig. 4G,H) (Sfeir et al, 2009; Lee et al, 2018). Of note, TRF1 depletion also elevated the frequency of outsider telomeres (Majerska et al, 2018), in which the telomeric FISH signal is detached from chromosome ends (Fig. EV3F), both at leading and lagging strands (Fig. EV3F,G). The outsider phenotype was observed upon TRF1 depletion in HeLa cells (Fernandes and Lingner, 2023) but not in mouse cells from which the gene had been deleted (Sfeir et al, 2009; Yang et al, 2022). In contrast, TERRA overexpression did not increase the frequency of outsider telomeres, which therefore represents a distinct feature of human cells lacking TRF1 (Fig. EV3G).

## BIR and PRIMPOL pathways act in parallel to repair TERRA R-loop-mediated telomere damage

According to the prevalent view, telomere fragility is a phenomenon that occurs subsequent to replication stress. However, the precise mechanisms of its formation remain unclear (Glousker and Lingner, 2021). It is believed that the fragile appearance of telomeres stems from repair of stalled or collapsed replication forks in which single-stranded DNA gaps are retained after repair and/or in which the repaired DNA is not fully condensed. TERRA R-loops may obstruct the replication machinery, leading to fork stalling and possibly fork collapse (García-Muse and Aguilera, 2019). One possible mechanism for restarting stalled forks might

involve repriming downstream of the R-loop, a process that can be facilitated by PRIMPOL. The PRIMPOL polymerase has, unlike other DNA polymerases, the unique ability to initiate replication without the need of an RNA primer (Bianchi et al, 2013). BIR is another repair pathway which we suspected to repair TERRA R-loop dependent damage in HeLa cells, as it had already been linked to TERRA-induced replication stress and telomere maintenance in ALT cells. BIR is dependent on the POLD3 accessory subunit of DNA polymerase δ (Costantino et al, 2014). Furthermore, POLD3 depletion in mouse embryonic fibroblasts (MEFs) suppressed telomere fragility caused by knocking out the BLM RecQ helicase (Yang et al, 2020).

We tested whether single depletion of either PRIMPOL (Fig. EV4A) or POLD3 (Fig. EV4B) affected TERRA-induced telomere fragility (Fig. 5A,B). However, no significant changes of fragility were seen in TERRA-overexpressing cells in which TERRA R-loops had become the major source of fragility (the increased fragility seen upon PRIMPOL or POLD3 in control cells may again, as discussed above for RAD51 and RAD51AP1, stem from roles of these factors in overcoming other replication stress sources). We therefore wondered whether the PRIMPOL and POLD3 pathways could compensate for each other and act in parallel pathways to overcome R-loop-dependent telomere replication stress. Thus, when one pathway was inhibited, cells might use the other one to repair the telomere DNA damage. To test this hypothesis, we co-depleted PRIMPOL and POLD3. Strikingly, co-depletion of PRIMPOL with POLD3 resulted in a notable suppression of TERRA-induced telomere fragility in TERRA-overexpressing cells (Fig. 5C). Of note, the reduction of fragility in TERRA-overexpressing cells was not linked to effects on cell cycle distribution or R-loop frequency (Fig. EV4C,D), confirming that PRIMPOL and POLD3 act downstream of TERRA R-loops. From the decrease in telomere fragility upon co-depletion of PRIMPOL and POLD3, we conclude that both factors are involved in the repair of TERRA-induced telomere damage and the generation of telomere fragility, acting in parallel pathways.

To further characterize the involved pathways, we depleted the fork reversal enzyme SMARCAL1 (Fig. EV4E) (Cox et al, 2016), depletion of which was reported to shift the balance toward PRIMPOL-dependent repriming (Quinet et al, 2020). We also depleted the fork cleavage enzyme MUS81 (Fig. EV4F) (Sobinoff et al, 2017), which is required for replication restart after R-loop mediated fork stalling (Chappidi et al, 2020). Similar to POLD3 depletion, single depletion of SMARCAL1 or MUS81 did not affect TERRA-induced fragility (Fig. 5D). However, co-depletion of either SMARCAL1 or MUS81 with PRIMPOL significantly reduced TERRA-induced telomere fragility (Fig. 5E). As seen for PRIMPOL and POLD3 depletion, this reduction was not due to decreased R-loop levels (Fig. EV4D). Thus, also SMARCAL1 and MUS81 act downstream of TERRA R-loops. Taken together, our findings highlight the critical roles of PRIMPOL-dependent repriming and POLD3-dependent BIR in telomeric DNA repair following replication stress.

## Telomere repair by conservative DNA synthesis

In PRIMPOL-mediated DNA repair, DNA synthesis should occur in a semiconservative manner using parental DNA as a template. In BIR, however, the broken telomeric DNA is elongated by conservative DNA synthesis. Upon strand invasion of the telomeric

3' overhang, Polδ initiates repair DNA synthesis. The newly synthesized leading strand subsequently serves as a template for Okazaki strand synthesis (lagging strand) initiated by the polymerase alpha (Polα)-primase complex. In the CO-FISH protocol, the newly synthesized DNA strands are degraded (Fig. 4F). Thus, telomeres synthesized by conservative telomeric DNA synthesis should appear as truncated telomeres upon CO-FISH but not FISH analysis (Fig. 6A,B). We tested this possibility with HeLa cells containing long telomeres of >30 kb. Consistent with the findings in HeLa cells with 10 kb telomeres, neither single depletion of PRIMPOL nor POLD3 significantly impacted TERRA-induced telomere fragility (Fig. EV5). The FISH analysis also demonstrated that telomere loss events were very rare in TERRA-overexpressing cells, independently of presence or absence of PRIMPOL and POLD3 (Fig. 6C). Significantly, however, in the CO-FISH analysis, telomere signal loss strongly increased upon TERRA overexpression, particularly at leading strand telomeres (Fig. 5C). In addition, telomere signal loss was augmented further upon PRIMPOL depletion, suggesting a shift towards BIR-mediated repair involving conservative DNA synthesis (Fig. 6C). POLD3 depletion showed a trend towards reduced telomere loss, which was, however, not statistically significant. The CO-FISH analysis indicates that ~5% of leading strand telomeres are repaired by conservative DNA synthesis (Fig. 6C; see leading strand telomeric loss) which is consistent with BIR. On the other hand, the CO-FISH analysis of telomere fragility indicated that due to overexpressed TERRA, ~7.5% of leading strand telomeres are repaired by semiconservative DNA replication (see Fig. 4H, subtract fragility in siNT in control to siNT in TERRA OE cells), which is consistent with PRIMPOL-dependent repair. We therefore estimate that both pathways are similarly contributing to overcoming TERRA-induced replication stress.

The CO-FISH analysis indicates that a significant portion of leading strand telomeres break off in the presence of TERRA R-loops but telomeres are resynthesized de novo by conservative DNA synthesis. Resynthesis must be extremely efficient as TERRA overexpression did nor induce a significant amount of telomere loss events as analyzed by FISH. At lagging strand telomeres, the CO-FISH analysis also indicates conservative DNA replication-mediated repair upon TERRA overexpression, which was, however, occurring at much lower levels (Fig. 6C).

Conservative DNA synthesis, which we observed in our CO-FISH analysis of TERRA overexpressing, is a hallmark of BIR (Saini et al, 2013). BIR occurs in G2 of the cell cycle (Kramara et al, 2018). In order to test if telomeric DNA synthesis occurred in TERRA-overexpressing cells in G2, we released cells for 4 h from a thymidine block induced G1-S arrest and treated them with a CDK1 inhibitor in order to arrest them in G2 (Silva et al, 2021; Zhang et al, 2019). To assess DNA synthesis during G2, EdU was added during the final 3 h of G2 arrest. Cells were then subjected to telomeric DNA FISH combined with the Click-iT reaction. EdU foci accumulated at telomeres in TERRA-overexpressing cells, indicating telomeric DNA synthesis in G2 (Fig. 6D–G). U2OS ALT cells which were used as a positive control also showed telomeric DNA synthesis in G2 at even higher levels. Altogether, the involvement of POLD3, SMARCAL1 and MUS81, the conservative telomeric DNA synthesis as well as the telomeric DNA synthesis in G2 corroborate the conclusion that TERRA overexpression induces BIR at telomeres.

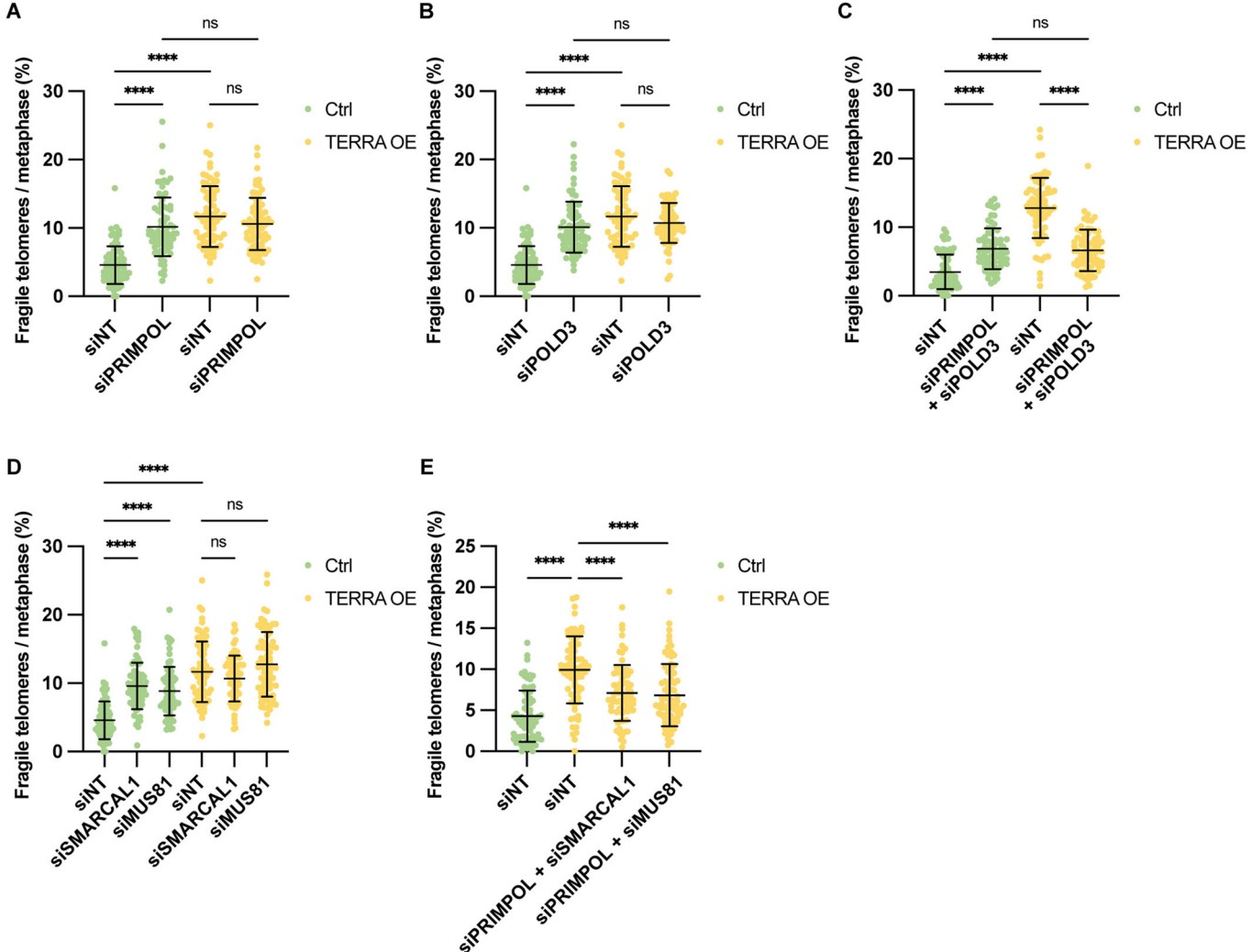

**Figure 5.  Repriming and BIR pathways act in parallel to repair TERRA R-loop-mediated telomere damage.**

(A) Quantification of telomere fragility upon depletion of PRIMPOL in control and TERRA-overexpressing HeLa cells. At least 25 metaphases were analyzed per condition per replicate, and three independent biological replicates were performed. Horizontal line and error bars represent mean ± s.d. Two-way analysis of variance (ANOVA) with Tukey's multiple comparisons test was applied. *P* values from left to right: ****$P < 0.0001$, ****$P < 0.0001$, ns $P = 0.9085$, ns $P = 0.3182$. (B) Quantification of telomere fragility upon depletion of POLD3 in control and TERRA-overexpressing HeLa cells. At least 25 metaphases were analyzed per condition per replicate, and three independent biological replicates were performed. Horizontal lines and error bars represent mean ± s.d. Two-way analysis of variance (ANOVA) with Tukey's multiple comparisons test was applied. *P* values from left to right: ****$P < 0.0001$, ****$P < 0.0001$, ns $P = 0.7053$, ns $P = 0.3600$. (C) Quantification of telomere fragility upon co-depletion of PRIMPOL and POLD3 in control and TERRA-overexpressing HeLa cells. At least 25 metaphases were analyzed per condition per replicate, and three independent biological replicates were performed. Horizontal lines and error bars represent mean ± s.d. Two-way analysis of variance (ANOVA) with Tukey's multiple comparisons test was applied. *P* values from left to right: ****$P < 0.0001$, ****$P < 0.0001$, ns $P = 0.9778$, ****$P < 0.0001$. (D) Quantification of telomere fragility upon depletion of PRIMPOL, SMARCAL1 or MUS81 in control and TERRA-overexpressing HeLa cells. At least 25 metaphases were analyzed per condition per replicate, and three independent biological replicates were performed. Horizontal lines and error bars represent mean ± s.d. Two-way analysis of variance (ANOVA) with Tukey's multiple comparisons test was applied. *P* values from left to right: ****$P < 0.0001$, ****$P < 0.0001$, ****$P < 0.0001$, ns $P = 0.5932$, ns $P = 0.4629$. (E) Quantification of telomere fragility upon co-depletion of PRIMPOL and SMARCAL1 or MUS81 in control and TERRA-overexpressing HeLa cells. At least 25 metaphases were analyzed per condition per replicate, and three independent biological replicates were performed. Horizontal lines and error bars represent mean ± s.d. One-way analysis of variance (ANOVA) with Dunnett's multiple comparisons test was applied. *P* values from left to right: ****$P < 0.0001$, ****$P < 0.0001$, ****$P < 0.0001$. Data from Figs. 4B,E and 5A,B,D stem from common experiments explaining the identity of the siNT control in these panels. Source data are available online for this figure.

Finally, we tested the importance of PRIMPOL and POLD3 for the survival of TERRA-overexpressing cells. Interestingly, TERRA overexpression in HeLa cells per se did not impair cell viability, indicating that the increased frequency of TIFs and fragile telomeres are well tolerated. Significantly, however, co-depletion of PRIMPOL and POLD3 impaired cell growth in TERRA-overexpressing cells much more severely than in control cells (Fig. 7A). This indicates that both repair pathways make crucial contributions to the survival of cells suffering from TERRA R-loop-induced telomere damage. Strikingly, co-depletion of PRIMPOL and POLD3 significantly impaired the cell growth of U2OS cells, whereas single depletion of either factor did not show a significant

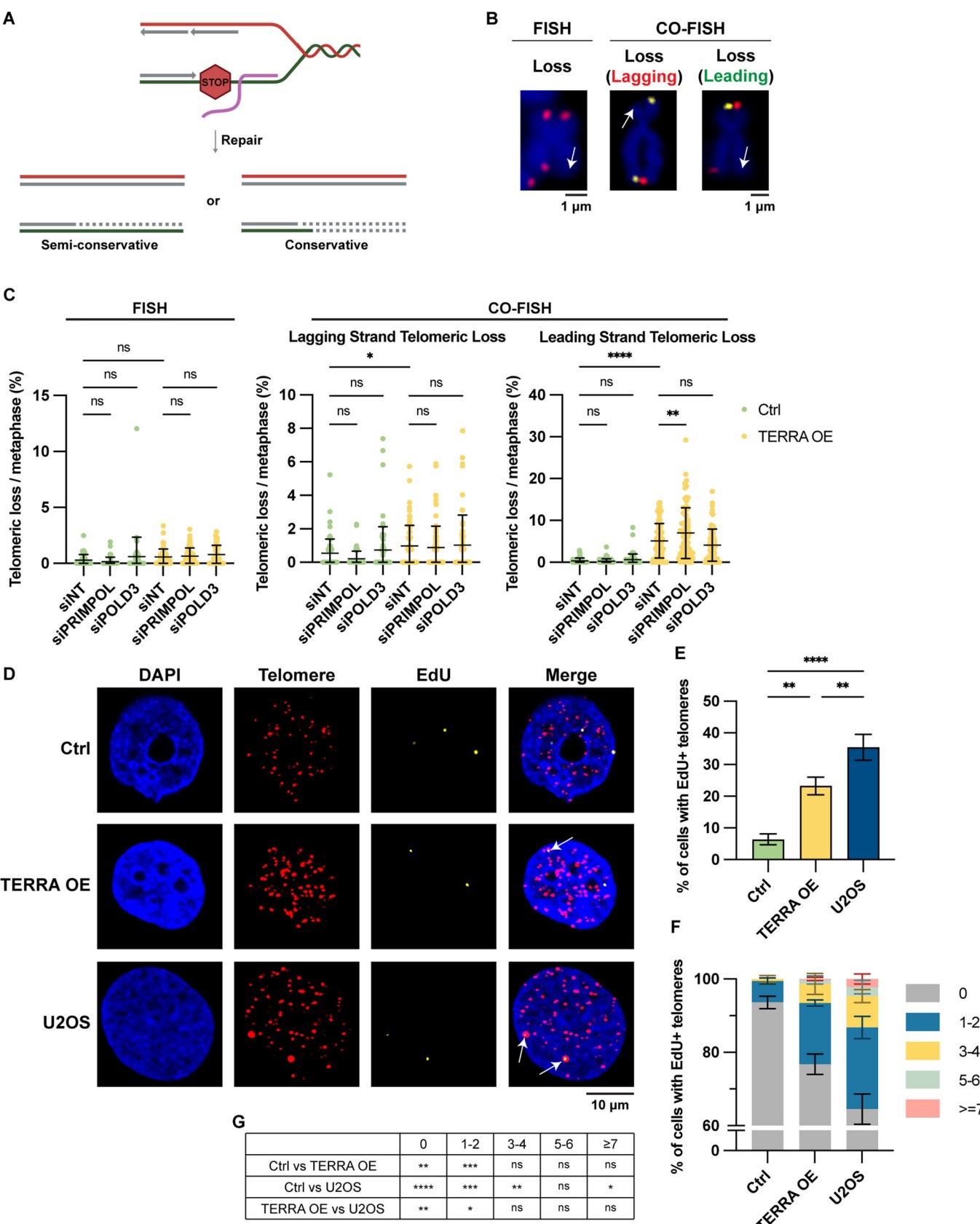

◀ **Figure 6. TERRA-induced DNA damage requires a conservative repair pathway.**

(A) Schematic representation of two possible scenarios that occur during DNA repair caused by TERRA R-loops. The red line represents the parental strand for lagging strand synthesis, while the green line represents the parental strand for leading strand synthesis. Solid gray lines indicate newly synthesized DNA during replication, and dashed gray lines represent newly synthesized DNA during repair. Both the solid and dashed gray lines are degraded and therefore not detected in CO-FISH experiments. (B) Representative images of telomeric loss by FISH (left) or CO-FISH (middle and right). Metaphase spreads were stained with telomeric (CCCTAA)$_3$-FISH probe (red) and DAPI (blue) (left), with TYE563-TeloC LNA probe (red), FAM-TeloG LNA probe (yellow), and DAPI (blue) (middle and right). (C) Quantification of telomeric loss upon depletion of PRIMPOL or POLD3 in control and TERRA-overexpressing HeLa cells with 30 kb telomeres. Metaphases from the same samples were analyzed by FISH (left) and CO-FISH (middle and right). At least 25 metaphases were analyzed per condition per replicate, and three independent biological replicates were performed. Horizontal lines and error bars represent mean ± s.d. Two-way analysis of variance (ANOVA) with Tukey's multiple comparisons test was applied. $P$ values from left to right: ns $P = 0.9488$, ns $P = 0.7600$, ns $P = 0.3306$, ns $P > 0.9999$, ns $P = 0.9956$ (left). ns $P = 0.2245$, ns $P = 0.6074$, *$P = 0.0296$, ns $P = 0.8659$, ns $P = 0.9807$ (middle). ns $P = 0.9811$, ns $P = 0.8554$, ****$P < 0.0001$, **$P = 0.0037$, ns $P = 0.1507$ (right). (D) Representative images showing EdU incorporation (yellow) at telomeres in control and TERRA-overexpressing HeLa cells. Click-iT detection was combined with telomeric (CCCTAA)$_3$-FISH (red) and DAPI staining (blue). White arrowheads EdU foci colocalizing with telomeres. (E) Quantification of cells exhibiting EdU-positive telomeres. Data represent mean ± s.d. from three independent biological replicates. One-way analysis of variance (ANOVA) with Tukey's multiple comparisons test was applied. $P$ values from left to right: **$P = 0.0012$, ****$P < 0.0001$, **$P = 0.0062$. (F) HeLa control, TERRA overexpression, and U2OS cells were categorized into five groups (0, 1–2, 3–4, 5–6, and ≥7) based on the number of EdU-positive telomeres per cell. Data represent mean ± s.d. from three independent biological replicates. (G) Statistical analysis of EdU-positive telomeres across control HeLa, TERRA-overexpressing HeLa, and U2OS cells. Data were analyzed using one-way analysis of variance (ANOVA) with Tukey's multiple comparisons test. $P$ values from left to right: **$P = 0.0012$, ***$P = 0.0010$, ns $P = 0.0658$, ns $P = 0.5794$, ns $P = 0.9257$ (Ctrl vs TERRA OE). ****$P < 0.0001$, ***$P = 0.0001$, **$P = 0.0066$, ns $P = 0.1116$, * $P = 0.0388$ (Ctrl vs U2OS). **$P = 0.0062$, *$P = 0.0273$, ns $P = 0.1849$, ns $P = 0.4046$, ns $P = 0.0616$ (TERRA OE vs U2OS). Source data are available online for this figure.

impact during the four-day time course (Fig. 7B). This further supports the critical interplay between PRIMPOL and POLD3 in supporting survival mechanisms required in cells with high TERRA levels.

## Discussion

The consequences and functions of TERRA R-loops have been studied in previous work mostly in the context of short and damaged telomeres during cellular senescence or at ALT telomeres, which contain a dramatically altered telomeric chromatin composition. In this paper, we investigate the formation and consequences of TERRA R-loops at healthy telomeres with normal length by inducing expression of endogenous TERRA with CRISPR activation. The strong increase in telomere fragility upon TERRA induction allowed us to distinguish the mechanisms related to TERRA R-loops from other sources of damage. Thus, important biological effects and functions at telomeres could be directly assigned to TERRA. Our work provides a framework explaining TERRA R-loop biology, which consists of multiple steps (Fig. 7C). First, elevated TERRA transcription leads to an increase in telomeric R-loops that not only form at the telomeres from which TERRA is expressed but also at telomeres in trans post transcription. Post-transcriptional TERRA R-loop formation requires both the RAD51 DNA recombinase and the RAD51-interacting protein RAD51AP1, which has been implicated in assisting HDR during synaptic complex formation and strand invasion (Pires et al, 2017). More recently, RAD51AP1 has also been demonstrated to be required for TERRA R-loop formation and telomere maintenance in ALT cancer cells (Barroso-González et al, 2019). Second, upon R-loop formation, TERRA R-loops interfere with the semiconservative DNA replication machinery leading to telomere damage and telomere fragility. Indeed, the telomere fragility was alleviated when TERRA R-loops did not form because of RAD51 or RAD51AP1 depletion or when they were destroyed by RNaseH1 overexpression. Third, the replication interference by TERRA R-loops leads to replication fork damage followed by DNA repair. Two pathways function in parallel to evade the TERRA R-loop-induced replication blockade. PRIMPOL, which can reprime and therefore restart DNA synthesis

downstream of obstacles, can overcome TERRA R-loop-mediated fork stalling. It remains uncertain if residual TERRA may serve as a primer for leading strand synthesis. Alternatively, POLD3-dependent repair occurring in G2 can kick in to repair TERRA R-loop-generated damage. The POLD3 accessory subunit of DNA polymerase δ is essential for BIR. Furthermore, the SMARCAL1 ATP-dependent strand annealing helicase is involved (Cox et al, 2016). SMARCAL1 catalyzes fork remodeling and/or fork reversal and, as such, appears to mediate repair stemming from R-loop-induced damage. The MUS81 nuclease is also required for the repair. MUS81 contributes to ALT-mediated DNA synthesis (Lu et al, 2024). MUS81 is part of a specialized endo-nucleolytic complex known as SMX (SLX1-4, MUS81-EME1, XPF-ERCC1) (Sobinoff et al, 2017) and may cleave impeded and rearranged recombination intermediates, possibly to generate telomeric DNA substrates for strand invasion. Consistent with TERRA R-loop-induced telomere cleavage, our CO-FISH analysis and its comparison to the analysis by FISH reveal de novo telomere synthesis by conservative replication. The CO-FISH data indicate that BIR-mediated conservative DNA synthesis is mostly engaged at leading strand telomeres in TERRA-overexpressing cells. The crucial importance of the POLD3 and PRIMPOL repair pathways for TERRA R-loop containing telomeres is apparent as concomitant impairment of POLD3 and PRIMPOL caused massive cell death in TERRA-overexpressing cells.

Overall, our results demonstrate that TERRA R-loops are sufficient to induce replication stress at telomeres, which is followed by telomere maintenance by BIR. This mechanism may become particularly important at short telomeres, especially in telomerase-negative cells, to prevent premature senescence (Graf et al, 2017; Sze et al, 2023). Indeed, we did not observe any changes in telomere length following TERRA induction in HeLa cells with 10 kb telomeres after several weeks of growth (Fig. EV5C). In telomerase-negative ALT cancer, the exact genetic and epigenetic alterations that lead to the activation of ALT-telomere maintenance mechanisms are unknown. We searched in TERRA-overexpressing HeLa cells for ALT-typical characteristics including enhanced sister chromatid exchange frequency, formation of C-circles and ALT-specific PML bodies (Fig. EV5D–F). However, none of these features was induced, indicating that TERRA has specific effects on telomere replication and repair. Thus, other

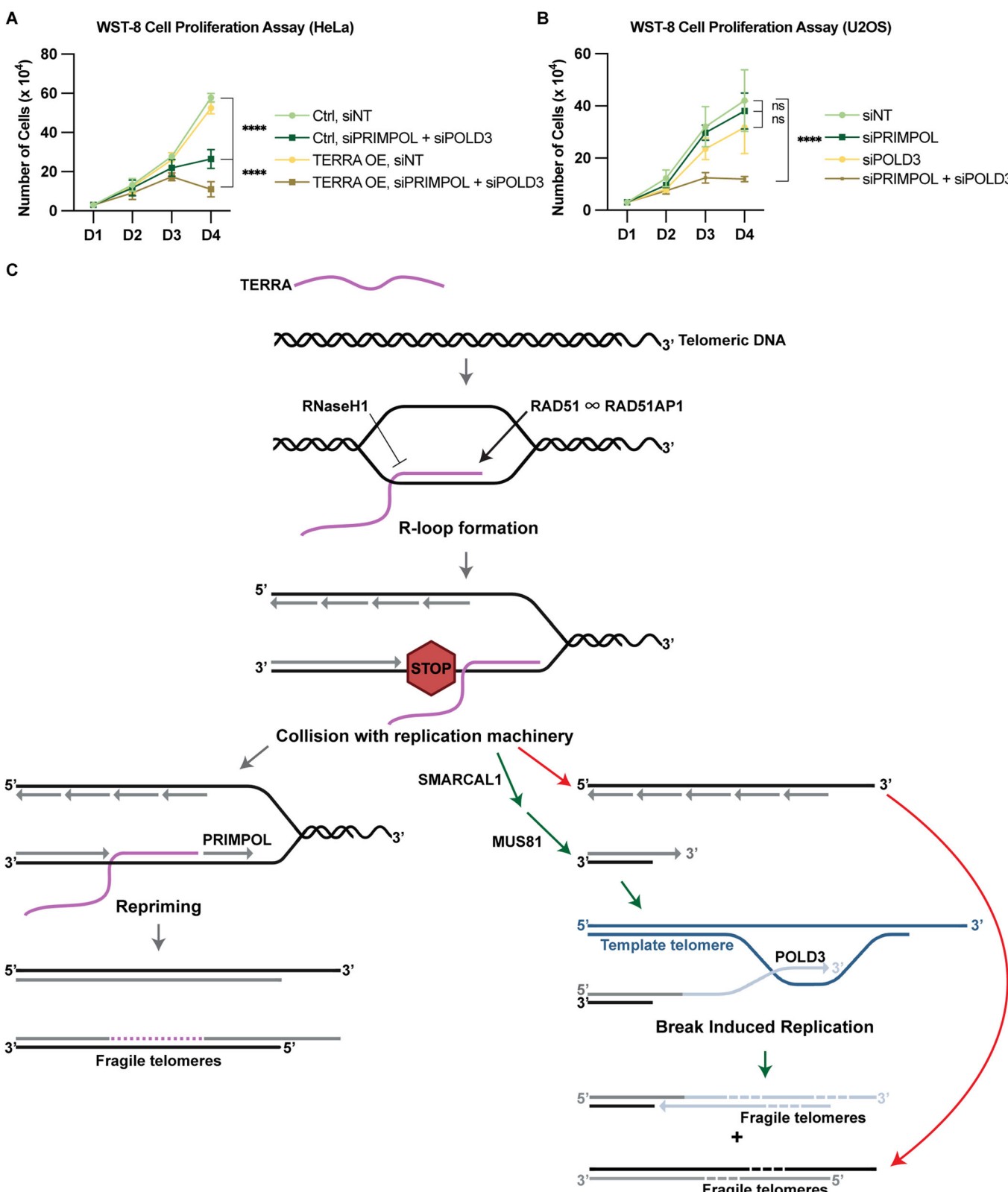

**Figure 7. Model for TERRA R-loop formation, replication interference and repair.**

(A) WST-8 cell proliferation assay upon co-depletion of PRIMPOL and POLD3 in control and TERRA-overexpressing HeLa cells. After seeding, cell viability was measured every 24 h. siRNA transfection was performed at 24 h. Data represent mean ± s.d. from three independent biological replicates. Two-way analysis of variance (ANOVA) with Tukey's multiple comparisons test was applied. $P$ values from upper to lower: ****$P < 0.0001$, ****$P < 0.0001$. (B) WST-8 cell proliferation assay upon single or co-depletion of PRIMPOL and/or POLD3 in U2OS cells. After seeding, cell viability was measured every 24 h. siRNA transfection was performed at 24 h. Data represent mean ± s.d. from three independent biological replicates. One-way analysis of variance (ANOVA) with Dunnett's multiple comparisons test was applied: $P$ values from upper to lower: ns $P = 0.8849$, ns $P = 0.3712$, ****$P < 0.0001$. (C) RAD51 and RAD51AP1 are both required for the formation of TERRA R-loops post transcription. TERRA R-loops can be destroyed by RNaseH1. In S phase, TERRA R-loops interfere with the DNA replication fork progression giving rise to telomere damage and repair. Two main pathways are involved in the repair of TERRA R-loop damaged DNA. Either PRIMPOL mediates fork restart downstream of TERRA R-loops or the truncated telomeric DNA is repaired by POLD3-dependent BIR. While the specific roles of SMARCAL1 and MUS81 in these processes have not been precisely defined, they function in parallel to PRIMPOL and may prepare the replication-impaired telomeric DNA for strand invasion and repair synthesis. Remaining gaps due to incomplete DNA repair synthesis may give rise to telomere fragility. Gray lines represent newly synthesized DNA by semiconservative DNA replication. Purple dashed lines denote gaps which may remain from TERRA R-loops. Light blue lines represent newly synthesized DNA via BIR, with gaps indicating regions of incomplete DNA synthesis. This model does only depict repair pathways at the leading strand telomeres. We speculate that elevated G-quadruplex formation at single-stranded G-rich strands, driven by TERRA R-loops may contribute to the enhanced fragility at the lagging strand. Source data are available online for this figure.

genetic or epigenetic changes are required to induce a complete ALT phenotype. It has been demonstrated that experimental depletion of histone chaperones ASF1a and ASF1b in human cells induced ALT features (O'Sullivan et al, 2014). ATRX-loss is also seen frequently in ALT cancer cells contributing to telomeric chromatin changes and TERRA expression (Heaphy et al, 2011). Our results indicate that increased TERRA R-loops are sufficient to induce BIR, suggesting that their establishment may represent a key event in ALT cancer cell evolution to overcome cell death during replicative cell crisis.

Intriguingly, during cell crisis, TERRA is upregulated at TRF2-depleted critically short telomeres acting as signaling molecule in the cytoplasm and triggering innate immune response-dependent autophagic cell death (Nassour et al, 2023). Thus, upregulated TERRA promotes cell death during crisis, acting as a tumor suppressor. TERRA upregulation during cell crisis, however, may become a two-edged sword and acquire oncogenic function when deviated into R-loops to initiate the BIR-mediated telomere maintenance mechanisms of ALT cells.

# Methods

### Reagents and tools table

| Reagent/ resource | Reference or source | Identifier or catalog number |
| --- | --- | --- |
| **Experimental models** | | |
| HeLa | Lingner Lab | N/A |
| HeLa Ctrl | This study | N/A |
| HeLa TERRA OE | This study | N/A |
| HeLa ST | Lingner Lab | Cristofari and Lingner, 2006 |
| HeLa ST Ctrl | This study | N/A |
| HeLa ST TERRA OE | This study | N/A |
| U2OS | Lingner Lab | N/A |
| **Recombinant DNA** | | |
| pcDNA6-myc-HIS | Lingner Lab | N/A |
| pcDNA6-RNAseH1-myc-His | Lingner Lab | N/A |

| Reagent/ resource | Reference or source | Identifier or catalog number |
| --- | --- | --- |
| pcDNA6-RNAseH1-D145N-myc-His | This study | N/A |
| lenti-dCAS-VP64_Blast | Konermann et al, 2015 | Addgene 61425 |
| gRNA_Cloning Vector | Mali et al, 2013 | Addgene 41824 |
| pLenti_gRNA 55_AAVS1 | This study | N/A |
| pLenti_gRNA 3_TERRA | This study | N/A |
| pLenti_gRNA 8_TERRA | This study | N/A |
| **Antibodies** | | |
| RAD51 | Santa Cruz | sc-8349 |
| RAD51AP1 | Proteintech | 11255-1-AP |
| TRF1 | Santa Cruz | sc-6165R |
| RNaseH1 | GeneTex | GTX117624 |
| PRIMPOL | Proteintech | 29824-1-AP |
| POLD3 | Abnova | H00010714-M01 |
| SMARCAL1 | Santa Cruz | sc-376377 |
| MUS81 | Abcam | ab14387 |
| Vinculin | Abcam | ab129002 |
| Tubulin | Sigma | T9026 |
| α-rabbit IgG (H + L) HRP-conjugated | Promega | W4011 |
| α-mouse IgG (H + L) HRP-conjugated | Promega | W4021 |
| S9.6 | Kerafast | ENH001 |
| 53BP1 | Novus Biologicals | NB100-304 |
| Phospho-RPA32 (Ser33) | Bethyl | A300-246A |
| PML | Abcam | ab96051 |
| goat anti-rabbit Alexa Fluor 633 | ThermoFisher | A-21070 |
| Acetyl-Histone H4 | Millipore | 06-866 |
| Normal mouse IgG | Santa Cruz | sc-2025 |

| Reagent/resource | Reference or source | Identifier or catalog number |
|---|---|---|
| Normal rabbit IgG | Santa Cruz | sc-2027 |
| **Oligonucleotides and other sequence-based reagents** | | |
| qPCR primer for 1q TERRA, forward | Feretzaki et al, 2020 | CAGCGTCGCAACTCAAATG |
| qPCR primer for 1q TERRA, reverse | Feretzaki et al, 2020 | CCCTCACCCTCCATGAGTAATA |
| qPCR primer for 2q TERRA, forward | Deng et al, 2012 | GCCTTGCCTTGGGAGAATCT |
| qPCR primer for 2q TERRA, reverse | Deng et al, 2012 | AAAGCGGGAAACGAAAAGC |
| qPCR primer for 7p TERRA, forward | Deng et al, 2012 | GGAGGCTGAGGCAGGAGAA |
| qPCR primer for 7p TERRA, reverse | Deng et al, 2012 | CAATCTCGGCTCACCACAATC |
| qPCR primer for 10q TERRA, forward | Deng et al, 2012 | GCCTTGCCTTGGGAGAATCT |
| qPCR primer for 10q TERRA, reverse | Deng et al, 2012 | AAAGCGGGAAACGAAAAGC |
| qPCR primer for 13q TERRA, forward | Feretzaki et al, 2020 | GCACTTGAACCCTGCAATACAG |
| qPCR primer for 13q TERRA, reverse | Feretzaki et al, 2020 | CCTGCGCACCGAGATTCT |
| qPCR primer for 15q TERRA, forward | Feretzaki et al, 2020 | CAGCGAGATTCTCCCAAGCTAAG |
| qPCR primer for 15q TERRA, reverse | Feretzaki et al, 2020 | AACCCTAACCACATGAGCAACG |
| qPCR primer for GAPDH, forward | Feretzaki et al, 2020 | AGCCACATCGCTCAGACAC |
| qPCR primer for GAPDH, reverse | Feretzaki et al, 2020 | GCCCAATACGACCAAATCC |
| RNaseH1 D145N point mutation primer, forward | This study | TACACTAATGGCTGCTGCTCCAGTAAT |
| RNaseH1 D145N point mutation primer, reverse | This study | GCAGCCATTAGTGTAGACGACGACGAA |
| TERRA reverse transcription primer | Feretzaki and Lingner, 2017 | CCCTAACCCTAACCCTAACCCTAACCCTAA |
| GAPDH reverse transcription primer | Feretzaki and Lingner, 2017 | GTGATCCGCCCGCCTCGGCCTCCCA |
| Guide RNA against AAVS1 locus (pLenti_gRNA 55_AAVS1) | This study | GGGGCCACTAGGGACAGGAT |
| Guide RNA against TERRA promoters (pLenti_gRNA 3_TERRA) | This study | GCAGGCGCAGAGAGGCG |
| Guide RNA against TERRA promoters (pLenti_gRNA 8_TERRA) | This study | CCGGCGCAGGCGCAGAG |

| Reagent/resource | Reference or source | Identifier or catalog number |
|---|---|---|
| siGENOME Non-Targeting siRNA | Dharmacon | D-001206-13 |
| siRAD51 | Dharmacon | M-003530-04 |
| siRAD51AP1 | Ouyang et al, 2021 | GCAGUGUAGCCAGUGAUUAUU |
| siRNaseH1 | Dharmacon | M-012595-00 |
| siTRF1 | Dharmacon | M-010542-02 |
| siPrimPol | Dharmacon | M-016804-00 |
| siPOLD3 | ThermoFisher | 4390824 |
| siSMARCAL1 | Dharmacon | J-013058-08 |
| siMUS81 | Lai et al, 2017 | CAGCCCUGGUGGAUCGAUUAUU |
| **Chemicals, enzymes, and other reagents** | | |
| Blasticidin | InvivoGen | ant-bl-1 |
| Puromycin | InvivoGen | ant-pr-1 |
| Hygromycin B Gold | InvivoGen | ant-hg-5 |
| Zeocin | Gibco | R25001 |
| VE-821 | Focus Biomolecules | FBM-10-5589 |
| KU-55933 | Selleck Chemicals | S1092 |
| Thymidine | Sigma-Aldrich | T1895-5G |
| RO-3306 | APExBIO | A8885 |
| Blocking reagent | Roche | 11096176001 |
| BrdU (5-Bromo-2'-deoxyuridine) | Sigma-Aldrich | B5002 |
| BrdC (5-Bromo-2'-deoxycytidine) | MP Biomedicals | MPB-210016690 |
| ChemiGlow | ProteinSimple | 60-12596-00 |
| DAPI | Sigma-Aldrich | D9542 |
| Demecolcine | Sigma-Aldrich | D7385 |
| Exonuclease III | New England Biolabs | M0206 |
| Hoechst 33258 | Invitrogen | H21491 |
| Lipofetamine 2000 | Invitrogen | 11668019 |
| Phase Lock Gel Heavy tubes (5PRIME) | VWR | 733-2478 |
| Power SYBR Green PCR Master Mix | ThermoFisher | 4368706 |
| ProLong Diamond Antifade mountant | ThermoFisher | P36965 |
| RNaseA | Promega | A7973 |
| RNaseH | Roche | 10786357001 |
| RNase, DNase-free | Roche | 11119915001 |
| RNasin Plus | Promega | N261B |
| SUPERase·In RNase Inhibitor | Invitrogen | AM2696 |
| SuperScript III Reverse Transcriptase | Invitrogen | 18080085 |
| TeloC LNA probe | Fernandes and Lingner, 2023 | Qiagen 5′ TYE563-CCC*TAACCC*TAACCC*TAA 3′ (*T) indicate LNA nucleotides |

| Reagent/resource | Reference or source | Identifier or catalog number |
|---|---|---|
| TeloG LNA probe | Fernandes and Lingner, 2023 | Qiagen 5′ 6-FAM-T*TAGGGT*TAGGGT*TAGGG 3′ (*T) indicate LNA nucleotides |
| Cell Counting Kit 8 (WST-8 / CCK8) | Abcam | ab228554 |
| CalPhos Mammalian Transfection Kit User Manual | Clontech | 631312 |
| Gibson Assembly Cloning Kit | New England Biolabs | E5510S |
| NucleoSpin RNA | Macherey-Nagel | 740955-250 |
| Click-iT™ EdU Cell Proliferation Kit for Imaging, Alexa Fluor™ 488 dye | ThermoFisher | C10337 |
| **Software** | | |
| AIDA image analyzer | Elysia Raytest | http://www.elysia-raytest.com/de/ |
| GraphPad Prism 10 | GraphPad | www.graphpad.com/ |
| BioRender.com | BioRender.com | https://www.biorender.com/ |
| Adobe Illustrator, 2023 | Adobe | https://www.adobe.com/products/illustrator |
| Fiji-ImageJ | NIH | https://fiji.sc |

## Cell culture

HeLa and HEK293T cells were cultured in Dulbecco's modified Eagle's medium (DMEM) (Gibco), supplemented with 10% fetal bovine serum (FBS) and 100 U/mL penicillin/streptomycin. Cells were maintained at 37 °C with 5% $CO_2$. The generation of HeLa cells with an average telomere length of 30 kb has been previously described (Cristofari and Lingner, 2006). The generation of TERRA-overexpressing cell lines is outlined below. Selection of dCas-VP64-expressing cells was carried out using 20 µg/mL blasticidin, gRNA-expressing cells were selected with 10 µg/mL puromycin, and cells expressing super-telomerase were selected with 250 µg/mL hygromycin B.

## Generation of TERRA-overexpressing cell lines

To induce endogenous TERRA expression, the lentiviral vector Lenti-dCas-VP64_Blast and gRNA-containing plasmids (see "Reagents and Tools Table") were introduced into HeLa cells. The gRNA-expressing plasmids contained a puromycin resistance gene. gRNAs were inserted into the modified vector via Gibson assembly (New England Biolabs), as described previously (Mali et al, 2013). The sequences of the gRNAs can be found in the "Reagents and Tools Table".

## Lentivirus production and cell transduction

Lentiviral vectors were generated as previously described (Wiznerowicz et al, 2007). HEK293T cells (11 million) were seeded into 15-cm dishes a day prior to transfection. A combination of 55 µg

pMD2-G, 102 µg PCMVR8.74 plasmids, and 157 µg transfer vector was transfected using the calcium phosphate method (CalPhos Mammalian Transfection Kit, Clontech). After overnight incubation, the medium was replaced, and the lentivirus-containing supernatant was collected over 24 h in two intervals. The supernatant was filtered (0.22 µm) and concentrated by ultracentrifugation at $50,000 \times g$ for 2 h at 16 °C. The viral pellet was resuspended in 500 µL PBS and stored at −80 °C. Lentivirus titration was performed in HCT116 cells using qPCR, as previously described (Barde et al, 2010).

## siRNA transfection

siRNAs were used at a final concentration of 10 nM for the calcium phosphate transfection method. The day before transfection, 200,000 cells were seeded in 6-well plates, ensuring 30–40% confluency the following day. A 100 µL transfection mix was prepared, consisting of 125 mM $CaCl_2$, 110 nM siRNA, and 1× HBSS (pH 7.4) containing 50 mM HEPES, 280 mM NaCl, 1.19 mM $Na_2HPO_4 \cdot 2H_2O$, and 10 mM KCl. The mix was incubated at RT for 1 min before being added to the cells in 1 mL of antibiotic-free DMEM supplemented with 10% FBS. Cells were harvested 72 h post-transfection.

## Plasmid transfection

Plasmid transfection was carried out using Lipofectamine 2000 (Invitrogen) according to the manufacturer's protocol with 4 µg of plasmid DNA. The day before transfection, 500,000 cells were seeded in 6-well plates to achieve 70–80% confluency by the time of transfection. A Lipofectamine pre-mix was prepared by combining 10 µL of Lipofectamine 2000 with 240 µL of Opti-MEM (ThermoFisher) and incubating the mixture at room temperature for 5 min. Simultaneously, a DNA mix was prepared by mixing 4 µg of plasmid DNA with 250 µL of Opti-MEM. The Lipofectamine pre-mix was then added to the DNA mix and incubated at room temperature for 10 min. Finally, 500 µL of the transfection mix was added to the cells. Cells were harvested 48 h after transfection.

## Western blotting

Cells were harvested by trypsinization and lysed in 2x Laemmli buffer. After heating at 95 °C for 5 min, proteins were separated by SDS-PAGE and transferred to nitrocellulose membranes. Membranes were blocked with 3% BSA in PBS-Tween and incubated with primary antibodies overnight at 4 °C. After washing, membranes were incubated with HRP-conjugated secondary antibodies, and signals were developed using chemiluminescence (ChemiGlow, BioTechne). Images were captured using a Fusion FX imaging system (Vilber).

## RNA purification, RNA dot blot, and RT-qPCR

RNA was extracted from 3 to 5 million cells using the NucleoSpin RNA kit (Macherey-Nagel). The process included two on-column and one in-solution rDNase digestion (Macherey-Nagel). For RNA dot blot analysis, the purified RNA was treated with DNase-free RNase (Roche) as a control.

For RNA dot blot, samples were denatured at 65 °C for 3 min and blotted onto a Hybond-XL membrane (Amersham) using a dot blot apparatus (Bio-Rad). The RNA was UV-crosslinked to the membrane, which was then blocked in Church buffer (0.5 M $NaHPO_4$, 1 mM EDTA, pH 8.0, 1% BSA, 7% SDS) for at least 1 h at 50 °C. The membrane was hybridized overnight at 50 °C with a $^{32}$P-radiolabeled telomeric probe in Church buffer. After hybridization, the membrane was washed twice with 2× SSC, 0.5% SDS, and twice with 1× SSC, 0.5% SDS, for 15 min per wash. The radioactive signal was detected using a Typhoon Biomolecular Imager (GE) after exposure to a phosphorimager screen. Following signal detection, the membrane was stripped by incubation in boiling 0.1× SSC with 1% SDS at 55 °C, repeated three times for 30 min each, and re-blocked in Church buffer for at least 1 h at 55 °C. A $^{32}$P-radiolabeled 18S rRNA oligonucleotide probe was then hybridized overnight at 55 °C, and the membrane was processed similarly as described for the telomeric probe.

For RT-qPCR analysis of TERRA levels, 3 μg of RNA was reverse transcribed using 200 U of SuperScript III reverse transcriptase (ThermoFisher Scientific), along with GAPDH and TERRA reverse primers (see "Reagents and Tools Table"). The reverse transcription was performed at 55 °C for 1 h, followed by heat inactivation at 70 °C for 15 min. Reactions without reverse transcriptase were included as controls. Five percent of the reaction was mixed with 2x Power SYBR Green PCR Master Mix (ThermoFisher Scientific) and 0.5 μM forward and reverse qPCR primers (see "Reagents and Tools Table"). qPCR was performed in a QuantStudio 6 Flex Real-Time PCR system (ThermoFisher Scientific) with the following cycling conditions: 95 °C for 10 min, followed by 40 cycles of 95 °C for 15 s, and 60 °C for 1 min for annealing and extension.

## Chromatin immunoprecipitation (ChIP)

Ten million cells were harvested and crosslinked with 1% formaldehyde at RT for 10 min. Crosslinking was quenched by adding 1 M Tris-HCl (final concentration 250 mM) and rotating the mixture for 5 min at room temperature. The cells were then washed three times with cold 1× PBS. The cell pellets were flash-frozen in liquid nitrogen and stored at −80 °C for further use. Frozen cell pellets were thawed on ice, resuspended in lysis buffer (1% SDS, 10 mM EDTA (pH 8.0), 50 mM Tris-HCl (pH 8.0), supplemented with cOmplete EDTA-free protease inhibitor (Roche)) and incubated on ice for 10 min. Samples were transferred to sonication vials with AFA fiber (Covaris), and sonicated with a Focused-Ultrasonicator (E220, Covaris) (10% duty factor, 140 W power, 200 cycles per burst, for 20 min), to shear DNA fragments of <500 bp. Sonicated samples were centrifuged at $21,000 \times g$ for 10 min. Supernatants were collected and diluted with 1:10 in IP buffer (1.1% Triton, 1.2 mM EDTA pH 8.0, 16.7 mM Tris-HCl pH 8.0, 300 mM NaCl, supplemented with 1 tablet of protease inhibitor). Antibodies were added to each sample and incubated overnight at 4 °C on a rotating wheel. Pre-blocked Sepharose G beads (incubated with ytRNA and IP buffer for 1 h) were added to each sample, followed by incubation at 4 °C for 4 h. After incubation, the beads were pelleted by centrifugation at $400 \times g$ for 2 min at 4 °C, and the supernatant was discarded. The samples were washed by rotating on a wheel at 4 °C for 5 min and centrifugation at $400 \times g$ for 2 min at 4 °C, sequentially with low-salt (0.1% SDS, 1% Triton, 2 mM EDTA pH 8.0, 20 mM Tris pH 8.0, 300 mM NaCl), high-salt (0.1% SDS, 1% Triton, 2 mM EDTA pH 8.0, 20 mM Tris pH 8.0, 500 mM NaCl), LiCl (250 mM LiCl, 1% NP-40, 1% Na-deoxycholate, 1 mM EDTA pH 8.0, 10 mM Tris pH 8.0), and TE (1 mM EDTA pH 8.0, 10 mM Tris pH 8.0) buffers. Washed beads and input samples were resuspended in crosslink reversal buffer (20 mM Tris-HCl pH 8.0, 0.1% SDS, 0.1 M $NaHCO_3$, 0.5 mM EDTA) supplemented with 10 μg/mL RNase (DNase-free (Roche)) and incubated at 65 °C overnight. DNA was purified using the NucleoSpin Gel and PCR Clean-up kit with NTB buffer (Macherey-Nagel) and eluted in 100 μL $H_2O$. Samples were then analyzed by dot blot and qPCR (see below).

## DNA:RNA hybrid immunoprecipitation (DRIP)

DRIP was carried out as described previously (Glousker et al, 2022). Briefly, cells were harvested 72 h after siRNA transfection or 48 h after plasmid transfection via trypsinization. In total, $1 \times 10^7$ cells per condition were resuspended in 175 μL of ice-cold RLN buffer (50 mM Tris-HCl, pH 8.0, 140 mM NaCl, 1.5 mM $MgCl_2$, 0.5% NP-40, 1 mM dithiothreitol (DTT), and 100 U/mL RNasin Plus (Promega)) and incubated on ice for 5 min. Nuclei were pelleted by centrifugation at 300 g for 2 min at 4 °C. The nuclear pellet was resuspended in 500 μL of RA1 buffer (NucleoSpin RNA purification kit, Macherey-Nagel) containing 1% 2-mercaptoethanol, then homogenized using a syringe with a $0.9 \times 40$ mm needle. The lysate was loaded into Phase Lock Gel Heavy tubes (5PRIME), mixed with 250 μL $H_2O$ and 750 μL phenol–chloroform-isoamyl alcohol (25:24:1, pH 7.8–8.2), and centrifuged at $13,000 \times g$ for 5 min at room temperature. The aqueous phase was transferred to a new tube, and 750 μL of cold isopropanol and NaCl (final concentration 50 mM) were added. After mixing, the samples were incubated on ice for 30 min to precipitate nucleic acids, followed by centrifugation at $10,000 \times g$ for 30 min at 4 °C. The resulting nucleic acid pellet was washed twice with 70% cold ethanol, air-dried, and dissolved in 130 μL of $H_2O$. The nucleic acids were then sonicated using a Covaris Focused-Ultrasonicator (E220, 10% duty factor, 140 W, 200 cycles per burst, 180 s) to generate fragments of 100–300 bp. The concentration of fragmented nucleic acids was measured using a NanoDrop spectrophotometer (ThermoFisher). For control, 10 μL of nucleic acids were digested with 10 μL of RNaseH (1 U/μL, Roche) or $H_2O$ in 150 μL of RNaseH buffer (20 mM HEPES-KOH pH 7.5, 50 mM NaCl, 10 mM $MgCl_2$, 1 mM DTT) and incubated at 37 °C for 90 min. The reaction was stopped by adding 2 μL of 0.5 M EDTA (pH 8.0). Nucleic acids were diluted 1:10 in DIP-1 buffer (10 mM HEPES-KOH pH 7.5, 275 mM NaCl, 0.1% SDS, 0.1% Na-deoxycholate, 1% Triton X-100) and pre-cleared with 40 μL of Sepharose Protein G beads (Cytiva) for 1 h at 4 °C. DRIP was carried out using 30 μg of nucleic acids from HeLa cells with 3 μg of the S9.6 antibody (Kerafast) or mouse IgG. After incubation overnight at 4 °C on a rotating wheel, samples were washed with buffer DIP-2 (50 mM HEPES-KOH pH 7.5, 140 mM NaCl, 1 mM EDTA pH 8.0, 1% Triton X-100, 0.1% Na-deoxycholate), buffer DIP-3 (50 mM HEPES-KOH pH 7.5, 500 mM NaCl, 1 mM EDTA pH 8.0, 1% Triton X-100, 0.1% Na-deoxycholate), buffer DIP-4 (10 mM Tris-HCl pH 8.0, 1 mM EDTA pH 8.0, 250 mM LiCl, 1% NP-40, 1% Na-deoxycholate), and TE buffer (10 mM Tris-HCl pH 8.0, 1 mM EDTA pH 8.0). Immuno-precipitated samples and inputs were eluted in 100 μL of crosslink

reversal buffer (20 mM Tris-HCl pH 8.0, 0.1% SDS, 0.1 M NaHCO₃, 0.5 mM EDTA) containing 10 µg/mL RNase (DNase-free (Roche)), and incubated at 65 °C overnight. DNA was purified using the QIAquick PCR Purification Kit (Qiagen) and eluted in 100 µL of H₂O. The purified samples were then analyzed by dot blot or qPCR.

## qPCR analysis of ChIP and DRIP samples

Each qPCR reaction consisted of 1 µL of purified DNA (from immunoprecipitated or diluted input samples as described), 5 µL of Power SYBR Green PCR Master Mix (ThermoFisher Scientific), 1 µM of forward and reverse primers (see "Reagents and Tools Table"), and H₂O to a final volume of 10 µL. Each sample was run in technical duplicates. The qPCR was performed with an initial denaturation at 95 °C for 10 min, followed by 40 cycles of 95 °C for 15 s and annealing/extension at 60 °C for 1 min, using a QuantStudio 6 Flex Real-Time PCR system (ThermoFisher Scientific). Serial dilutions (with factors of 5 and 50) of each input sample were included to generate a standard curve via regression analysis. Based on the input equation, the corresponding immunoprecipitation values were calculated as a percentage of the input.

## Dot blot analysis of ChIP and DRIP samples

Input samples (from ChIP and DRIP experiments) were appropriately diluted to ensure the DNA concentration in each immunoprecipitated sample remained within the assay's linear dynamic range. Both purified DNA from diluted input and immunoprecipitated samples were heated at 95 °C for 5 min, then immediately cooled on ice. The samples were then blotted onto a Hybond-XL membrane (Amersham) using a dot blot apparatus (Bio-Rad) and UV-crosslinked to the membrane. The membrane was denatured in a solution of 0.5 M NaOH and 1.5 M NaCl for 15 min at room temperature on a shaker, followed by neutralization in 0.5 M Tris-Cl (pH 7.0) and 1.5 M NaCl for 10 min. The membrane was blocked in Church buffer (0.5 M NaHPO₄, 1 mM EDTA, pH 8.0, 1% BSA, 7% SDS) at 65 °C for at least 1 h. Hybridization was performed overnight at 65 °C with a ³²P-radiolabeled TeloC probe in Church buffer. The membrane was then washed three times for 30 min at 65 °C in 1× SSC containing 0.5% SDS, and exposed to a phosphorimager screen. The radioactive signal was detected using a Typhoon Biomolecular Imager (GE).

## Immunofluorescence and telomeric fluorescence in situ hybridization (IF-FISH)

For zeocin treatment, cells were incubated with 100 µg/mL of zeocin for 3 h. For ATR inhibitor (VE-821) and ATM inhibitor (KU-55933) treatment, cells were incubated with 10 µM of each inhibitor for 24 h. Cells were cultured on round coverslips, washed twice with PBS, and fixed in 4% paraformaldehyde for 10 min at room temperature. After fixation, cells were permeabilized with a detergent solution (0.1% Triton X-100, 0.02% SDS in PBS) and blocked with 2% BSA in PBS for 10 min. Primary antibodies (1:2000 for 53BP1 and 1:500 for PML) were applied in a blocking solution (10% normal goat serum, 2% BSA in PBS) and incubated overnight at 4 °C. Secondary antibodies (1:600) were applied for 30 min at room temperature after multiple washes. Coverslips were

then fixed again in 4% paraformaldehyde, washed, and dehydrated with a graded ethanol series (70%, 95%, and 100%). For FISH, hybridization was performed using 100 nM Cy3-[CCCTAA]3 PNA probe in a hybridization mix (10 mM Tris, 70% formamide, 0.5% blocking reagent) at 80 °C for 3 min, followed by incubation at room temperature for 3 h. Post-hybridization washes were performed twice with wash buffer 1 (10 mM Tris, 70% formamide) and three times with wash buffer 2 (0.1 M Tris, 0.15 M NaCl, 0.08% Tween-20). 100 ng/mL of DAPI was added to the second wash step of wash buffer 2. Coverslips were mounted with Vectashield, and images were captured with a Leica SP8 confocal microscope equipped with a 63 ×/ 1.40 oil objective and a DFC 7000 GT camera.

For pRPA IF-FISH, cells were fixed with 4% formaldehyde, permeabilized with CSK buffer (10 mM PIPES pH 7.0, 100 mM NaCl, 300 mM sucrose, 3 mM MgCl₂, 0.5% Triton X-100) and blocked in PBS containing 0.5% BSA and 0.1% Tween-20. Primary antibody (1:2000) and secondary antibody (1:1000) were diluted in the PBS blocking solution. Slides were washed with PBS containing 0.1% Tween-20.

## Telomeric FISH on metaphase chromosomes

Telomeric FISH was performed as previously described (Fernandes and Lingner, 2023). Cells were treated with 0.05 µg/mL demecolcine for 2 h before being harvested and subjected to hypotonic treatment (0.056 M KCl) at 37 °C for 7 min. After fixation in cold methanol acetic acid (3:1) overnight, cells were spread onto slides and processed for FISH staining, as described above, using a 100 nM Cy3-[CCCTAA]3 PNA probe. Images were captured using a Zeiss Axioplan or Leica SP8 confocal microscope with a Upright Zeiss Axioplan equipped with a 100x/1.40 oil objective.

## Chromosome orientation (CO)-FISH on metaphase chromosomes

CO-FISH was carried out as previously described (Fernandes and Lingner, 2023). Cells were incubated with BrdU/BrdC (3:1, final concentration of 10 µM) for 15 h and treated with 0.1 µg/mL demecolcine for 2 h. After hypotonic treatment, cells were fixed and spread onto glass slides. After rehydration, cells were treated with RNaseA (Promega), stained with Hoechst 33258, and exposed to UV light. Following exonuclease digestion and paraformaldehyde fixation, hybridization was performed using TYE563-TeloC LNA and 6-FAM-TeloG LNA probes (Qiagen). Post-hybridization washes were performed, and slides were mounted with ProLong Diamond Antifade mountant. Images were acquired using a Leica SP8 confocal microscope equipped with a ×63/1.40 oil objective and a DFC 7000 GT camera.

## Telomeric FISH and EdU detection

For cell cycle synchronization, cells cultured on round coverslips were incubated in medium containing 2 mM thymidine (Sigma-Aldrich) for 21 h. Cells were released into fresh medium for 4 h, followed by treatment with 10 µM RO-3306 (APExBIO) for 18 h. Subsequently, 10 µM EdU (ThermoFisher) was added, and cells were incubated for an additional 3 h. Cells were fixed in 4% paraformaldehyde for 10 min at room temperature, washed with

PBS, and permeabilized with a detergent solution (0.1% Triton X-100, 0.02% SDS in PBS). After three washes with PBS, cells were dehydrated through a graded ethanol series (70%, 95%, and 100%). Telomeric FISH was performed as described above, except that DAPI was omitted from the second wash step of wash buffer 2. Following ethanol dehydration, EdU incorporation was detected using the Click-iT EdU Alexa Fluor 488 Imaging Kit (Thermo-Fisher) according to the manufacturer's instructions. Slides were washed three times with PBS, including 100 ng/mL DAPI in the second wash. Coverslips were mounted with Vectashield, and images were captured with a Leica SP8 confocal microscope equipped with a ×63/1.40 oil objective and a DFC 7000 GT camera.

## WST-8 cell proliferation assay

The WST-8 cell proliferation assay was conducted according to the manufacturer's protocol (Abcam, Ab228554). Briefly, 15,000 cells were seeded in 24-well plates. At 24, 48, 72, and 96 h (corresponding to days 2, 3, 4, and 5), the culture media were replaced with 500 μL of WST-8 solution (450 μL of pre-warmed fresh media mixed with 50 μL of WST-8 reagent). After incubating the plates at 37 °C for 1 h, 200 μL of the supernatant was transferred to a 96-well plate (Falcon, 351172), and the absorbance was measured at 460 nm using a plate reader. siRNA transfection was performed on day 2 using the calcium phosphate transfection method.

## Telomere restriction fragment (TRF) analysis

TRF analysis was performed as described (Glousker et al, 2020). Briefly, genomic DNA from 3 to 5 million cells was extracted and digested with 30 U of RsaI and 50 U of HindIII. DNA was separated on a 0.8% agarose gel, denatured (0.5 M NaOH, 1.5 M NaCl), neutralized (1.5 M NaCl, 0.5 M Tris-HCl pH 7.5) and hybridized overnight with a $^{32}$P-labeled telomeric probe in Church buffer. Radioactive signals were detected with an Amersham Typhoon imager and quantified using AIDA software (v4.06.034).

## C-circle assay

The C-circle assay was performed as previously described (Glousker et al, 2020). Briefly, genomic DNA was extracted with phenol–chloroform–isoamyl alcohol (25:24:1) and digested with HinfI and RsaI (NEB) in CutSmart buffer at 37 °C overnight. In all, 30 ng of digested DNA was incubated with or without 7.5 U Phi29 DNA polymerase in the presence of 1 mM dATP, dTTP, and dGTP at 30 °C for 8 h, followed by heat inactivation at 65 °C. Products were blotted onto a Hybond-XL membrane using a dot blot apparatus, UV-crosslinked, and hybridized with a $^{32}$P-labeled TeloC probe in Church buffer at 42 °C overnight. Signals were detected using a Typhoon Biomolecular Imager (GE).

## Software

Image analysis was done using ImageJ, dot blots were analyzed with Aida Image Analyzer, and statistical analyses and graph preparation were performed using GraphPad Prism. Illustrations were created using Adobe Illustrator 2023 and BioRender.com.

## Data availability

This study includes no data deposited in external repositories.

The source data of this paper are collected in the following database record: biostudies:S-SCDT-10_1038-S44318-025-00502-4.

## Peer review information

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

## Acknowledgements

The authors thank Eftychia Kyriacou, Satyajeet Rao, and Galina Glousker for comments on the manuscript and Samah Matmati for technical advice. The authors acknowledge the support provided by the Bioimaging and Optics Platform, the Gene Expression Core Facility and the Flow Cytometry Core Facility at the School of Life Sciences of EPFL. This work was supported by the Swiss National Science Foundation (SNSF) [310030_214833] and the SNSF-funded National Centre of Competence in Research RNA and Disease Network [205601]. PRN was recipient of PhD fellowship from the Boehringer Ingelheim Fonds.

## Author contributions

**Suna In**: Conceptualization; Data curation; Formal analysis; Funding acquisition; Validation; Investigation; Methodology; Visualization; Writing—original draft; Writing—review and editing. **Patricia Renck Nunes**: Methodology; Visualization; Writing—review and editing. **Rita Valador Fernandes**: Methodology; Visualization. **Joachim Lingner**: Conceptualization; Formal analysis; Supervision; Funding acquisition; Writing—original draft; Project administration; Writing—review and editing.

Source data underlying figure panels in this paper may have individual authorship assigned. Where available, figure panel/source data authorship is listed in the following database record: biostudies:S-SCDT-10_1038-S44318-025-00502-4.

## Disclosure and competing interests statement

The authors declare no competing interests.

# Expanded View Figures

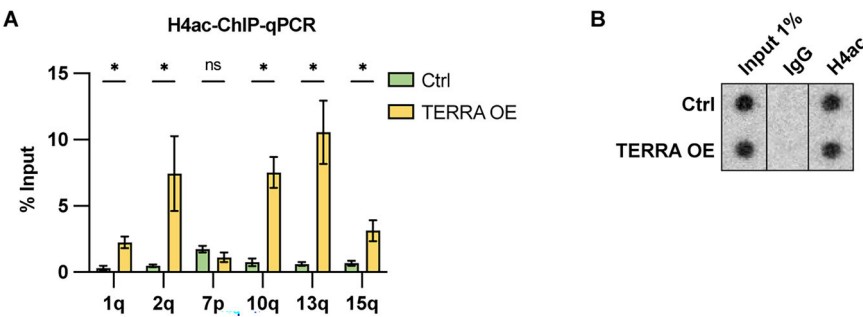

**Figure EV1.  Induction of endogenous TERRA by modified CRISPR-Cas9 system, related to Fig. 1.**

(A, B) ChIP assay using H4 acetylation antibody. ChIP samples and inputs were treated with RNase (DNase-free) and analyzed by qPCR with indicated subtelomeric primers (A) or by DNA dot blot with a $^{32}$P-radiolabeled telomeric probe (B). Multiple unpaired *t* test was applied. Data represent mean ± s.d. from three independent biological replicates. *P* values from left to right: *P = 0.0166, *P = 0.0269, ns P = 0.3148, *P = 0.0290, *P = 0.0163, *P = 0.0271. Source data are available online for this figure.

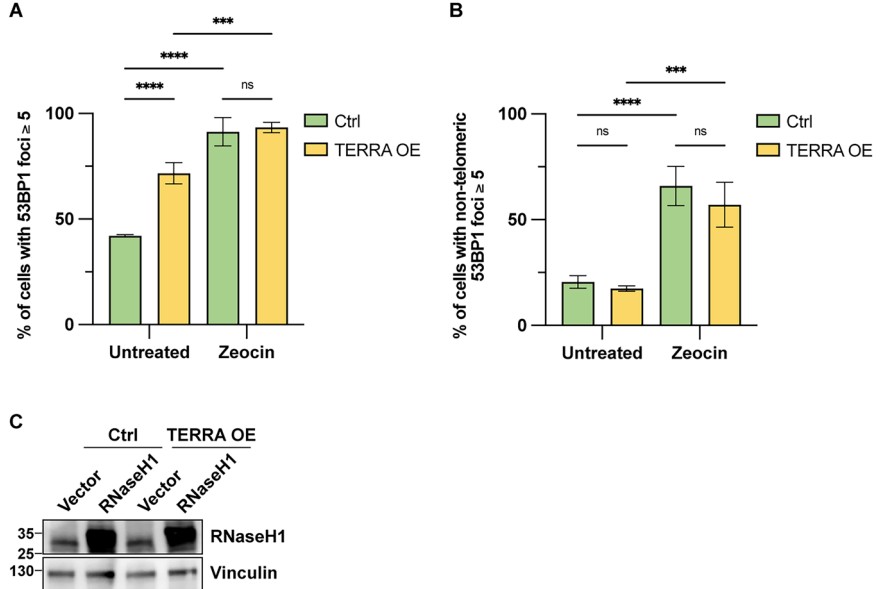

**Figure EV2. DNA damage upon TERRA overexpression and zeocin treatment, related to Fig. 3.**

(A) Quantification of the number of cells with ≥5 53BP1 foci. Data represent mean ± s.d. from three independent biological replicates. Two-way analysis of variance (ANOVA) with uncorrected Fisher's least significant difference (LSD) test was applied. *P* values from left to right: ****$P < 0.0001$, ****$P < 0.0001$, ***$P = 0.0003$, ns $P = 0.5932$. (B) Quantification of the number of cells with ≥5 53BP1 foci that are not colocalizing with telomeres. Data represent mean ± s.d. from three independent biological replicates. Two-way analysis of variance (ANOVA) with uncorrected Fisher's least significant difference (LSD) test was applied. *P* values from left to right: ns $P = 0.6179$, ****$P < 0.0001$, ***$P = 0.0002$, ns $P = 0.1704$. (C) Western blot analysis upon ectopic expression of RNaseH1 in control and TERRA-overexpressing HeLa cells. Source data are available online for this figure.

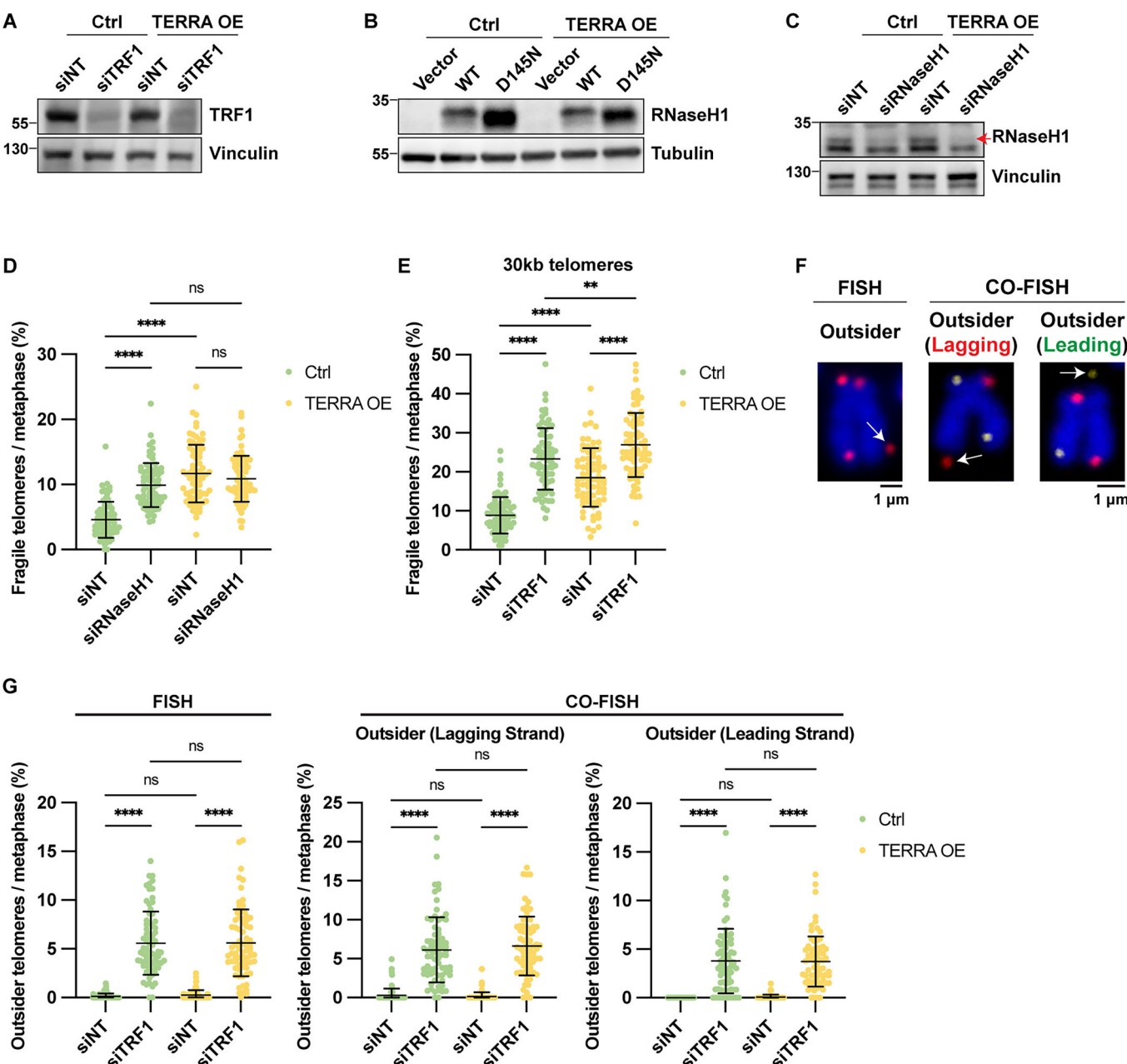

**Figure EV3.** **Fragile and outsider telomeres, related to Fig. 4.**

(**A**) Western blot analysis upon depletion of TRF1 in control and TERRA-overexpressing HeLa cells. (**B**) Western blot analysis upon ectopic expression of RNaseH1 WT or D145N mutant in control and TERRA overexpressing HeLa cells. (**C**) Western blot analysis upon depletion of RNaseH1 in control and TERRA-overexpressing HeLa cells. (**D**) Quantification of telomere fragility upon depletion of RNaseH1 in control and TERRA overexpressing HeLa cells. At least 25 metaphases were analyzed per condition per replicate, and three independent biological replicates were performed. Horizontal lines and error bars represent mean ± s.d. Two-way analysis of variance (ANOVA) with Tukey's multiple comparisons test was applied. *P* values from left to right: ****$P < 0.0001$, ****$P < 0.0001$, ns $P = 0.3221$, ns $P = 0.5391$. (**E**) Quantification of telomere fragility upon depletion of TRF1 in control and TERRA-overexpressing HeLa cells with 30 kb telomeres. At least 25 metaphases were analyzed per condition per replicate, and three independent biological replicates were performed. Horizontal lines and error bars represent mean ± s.d. Two-way analysis of variance (ANOVA) with Tukey's multiple comparisons test was applied. *P* values from left to right: ****$P < 0.0001$, ****$P < 0.0001$, ****$P < 0.0001$, **$P = 0.0012$. (**F**) Representative images of outsider telomeres by FISH (left) or CO-FISH (middle and right). Metaphase spreads were stained with telomeric (CCCTAA)$_3$-FISH probe (red) and DAPI (blue) (left), with TYE563-TeloC LNA probe (red), FAM-TeloG LNA probe (yellow), and DAPI (blue) (middle and right). (**G**) Quantification of outsider telomeres upon depletion of TRF1 in control and TERRA-overexpressing HeLa cells with 30 kb telomeres. Metaphases from the same samples were analyzed by FISH (left) and CO-FISH (middle and right). At least 25 metaphases were analyzed per condition per replicate, and three independent biological replicates were performed. Horizontal lines and error bars represent mean ± s.d. Two-way analysis of variance (ANOVA) with Tukey's multiple comparisons test was applied. *P* values from left to right: ****$P < 0.0001$, ns $P = 0.9852$, ns $P = 0.9997$, ****$P < 0.0001$ (left). ****$P < 0.0001$, ns $P = 0.9907$, ns $P = 0.7167$, ****$P < 0.0001$ (middle). ****$P < 0.0001$, ns $P = 0.9971$, ns $P = 0.9987$, ****$P < 0.0001$ (right). Source data are available online for this figure.

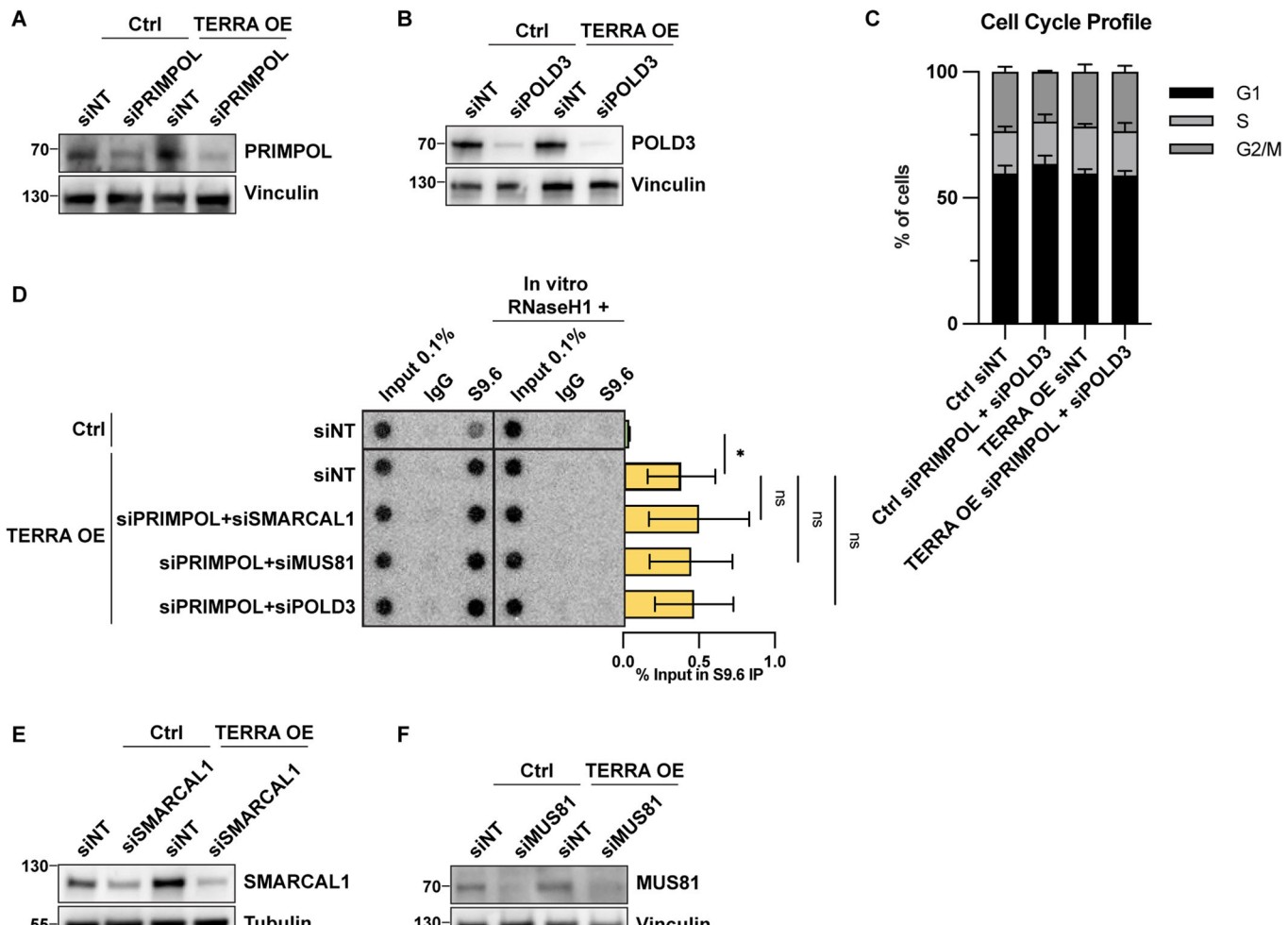

**Figure EV4. Western blot analyses of depletion of proteins involved in telomere fragility, related to Fig. 5.**

(A) Western blot analysis upon depletion of PRIMPOL in control and TERRA-overexpressing HeLa cells. (B) Western blot analysis upon depletion of POLD3 in control and TERRA-overexpressing HeLa cells. (C) Cell cycle profiles upon PRIMPOL and POLD3 depletion. DNA content was analyzed by flow cytometry analysis of fixed DAPI-stained cells at day 4 of the time course (last time point). Data represent mean ± s.d. from three independent biological replicates. (D) DRIP assay using S9.6 antibody. DRIP samples and inputs were treated with RNase (DNase-free) and analyzed by DNA dot blot with a $^{32}$P-radiolabeled telomeric probe. As a negative control, samples were treated in vitro with RNaseH1 prior to immunoprecipitation and analyzed in parallel. Data represent mean ± s.d. from three independent biological replicates. One-way analysis of variance (ANOVA) with Šídák's multiple comparisons test was applied. P values from upper to lower: *$P = 0.0389$, ns $P = 0.9668$, ns $P = 0.9965$, ns $P = 0.9899$. (E) Western blot analysis upon depletion of SMARCAL1 in control and TERRA-overexpressing HeLa cells. (F) Western blot analysis upon depletion of MUS81 in control and TERRA-overexpressing HeLa cells. Source data are available online for this figure.

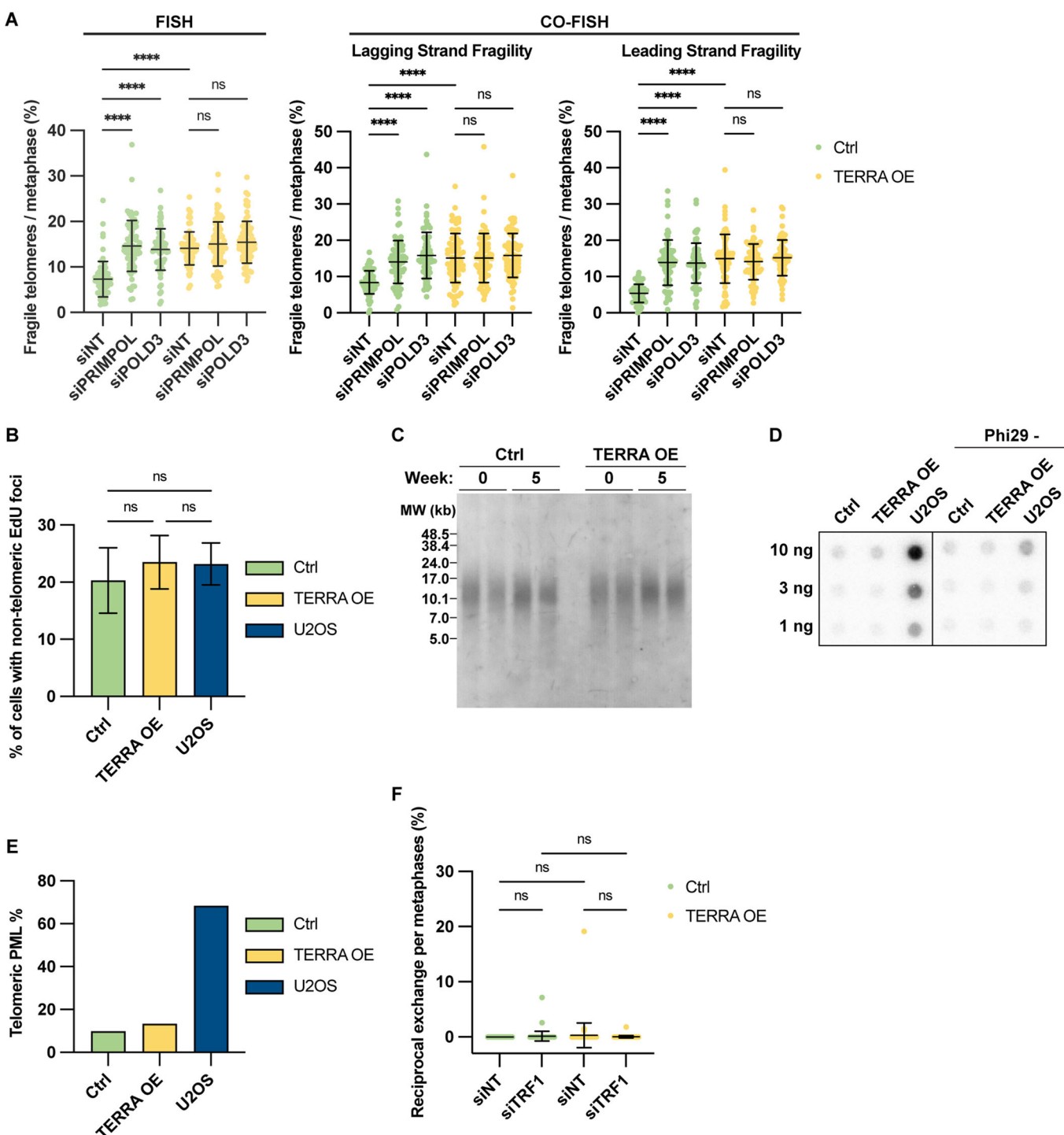

◀ **Figure EV5. Impacts of depletion of PRIMPOL, POLD3, SMARCAL1, and MUS81, related to Fig. 6.**

(A) Quantification of telomere fragility upon depletion of PRIMPOL or POLD3 in control and TERRA-overexpressing HeLa cells with 30 kb telomeres. Metaphases from the same samples were analyzed by FISH (left) and CO-FISH (middle and right). At least 25 metaphases were analyzed per condition per replicate, and three independent biological replicates were performed. Horizontal lines and error bars represent mean ± s.d. Two-way analysis of variance (ANOVA) with Tukey's multiple comparisons test was applied. $P$ values from left to right: ****$P < 0.0001$, ****$P < 0.0001$, ****$P < 0.0001$, ns $P = 0.3957$, ns $P = 0.1627$ (left). ****$P < 0.0001$, ****$P < 0.0001$, ****$P < 0.0001$, ns $P = 0.9998$, ns $P = 0.7558$ (middle). ****$P < 0.0001$, ****$P < 0.0001$, ****$P < 0.0001$, ns $P = 0.5791$, ns $P = 0.9580$ (right). (B) Quantification of cells exhibiting non-telomeric EdU foci. Data represent mean ± s.d. from three independent biological replicates. One-way analysis of variance (ANOVA) with Tukey's multiple comparisons test was applied. $p$ values from left to right: ns $P = 0.7029$, ns $P = 0.7485$, ns $P = 0.9964$. (C) Telomere restriction fragment (TRF) analysis of control and TERRA-overexpressing HeLa cells at baseline ("Week 0," immediately before gRNA transduction) and after 5 weeks ("Week 5"). (D) Phi29 C-circle assay performed on 10 ng, 3 ng, and 1 ng of DNA from control and TERRA-overexpressing HeLa cells, as well as U2OS cells. Amplification products were analyzed by dot blot using a $^{32}$P-labeled C-rich telomeric probe. (E) Quantification of PML bodies colocalizing with telomeres. Data represent one biological replicate. (F) Quantification of reciprocal telomeric sister chromatid exchange, as a percentage of events per metaphase spread. At least 25 metaphases were analyzed per condition per replicate, and three independent biological replicates were performed. Horizontal lines and error bars represent mean ± s.d. Two-way analysis of variance (ANOVA) with Tukey's multiple comparisons test was applied: ns indicates non-significance ($P > 0.05$). $P$ values from left to right: ns $P = 0.9110$, ns $P = 0.4455$, ns $P = 0.9490$, ns $P = 0.5211$. Source data are available online for this figure.

