## [Peer Review File · The EMBO Journal]

TERRA R-loops trigger a switch in telomere maintenance towards break-induced replication and PRIMPOL-dependent repair

Suna In, Patricia Renck Nunes, Rita Valador Fernandes, and Joachim Lingner

Corresponding author(s): Joachim Lingner (joachim.lingner@epfl.ch)

Review Timeline:

Submission Date:	7th Jan 25
Editorial Decision:	25th Feb 25
Revision Received:	27th May 25
Accepted:	23rd Jun 25

Editor: Hartmut Vodermaier

Transaction Report:

Dr. Joachim Lingner
EPFL
ISREC
Station 19
Lausanne 1015
Switzerland

25th Feb 2025

Re: EMBOJ-2025-120125
TERRA R-loops trigger a switch in telomere maintenance towards break-induced replication and PrimPol-dependent repair

Dear Joachim,

Thank you for submitting your study on the effects of inducible endogenous TERRA expression in non-ALT cells. I apologize once more for the delay in getting back to you with a decision. Your study has been seen by three expert referees (see reports below), who all appreciated the newly established technology for TERRA induction, but returned somewhat divergent views on the perceived significance of the results obtained with it: while referee 3 considers several findings interesting and important, referee 1 does not feel that they are sufficiently surprising to substantially advance our understanding. Finally, referee 2 is more ambivalent but also retains a number of substantive concerns whose adequate addressing would be required before possible publication.

Faced with these equivocal opinions, I would on balance still give you an opportunity to respond to the reports by way of a revised version of the manuscript. However, I should emphasize that further consideration for The EMBO Journal would require not just comprehensive addressing of the various specific issues, but ideally also extending the use of the new method to derive some deeper/novel insight (e.g. along the lines suggested by ref 2 point 13, or ref 1's request for clarification of the proposed model). I would therefore encourage you to contact me with a revision plan and preliminary point-by-point response already during the early stages of your revision work, so that we could discuss (via email or Zoom call) if and how this could be achieved. We would of course also be open to extension of the default three-months revision period if needed; our 'scooping protection' (meaning that competing work appearing elsewhere in the meantime will not affect our considerations of your study) remaining valid also throughout such an extension.

Detailed information on preparing, formatting and uploading a revised manuscript can be found below and in our Guide to Authors. Thank you again for the opportunity to consider this work for The EMBO Journal, and I look forward to hearing from you in due time.

With kind regards,

Hartmut

- size of the scale bars that are mandatory for all micrograph panels
- the statistical test used to generate error bars and P-values
- the type error bars (e.g., S.E.M., S.D.)
- the number (n) and nature (biological or technical replicate) of independent experiments underlying each data point

- Figures may not include error bars for experiments with $n < 3$; scatter plots showing individual data points should be used instead.

- 3) Revised manuscript text (including main tables, and figure legends for main and EV figures) has to be submitted as editable text file (e.g., .docx format). We encourage highlighting of changes (e.g., via text color) for the referees' reference.
- 4) Each main and each Expanded View (EV) figure should be uploaded as individual production-quality files (preferably in .eps, .tif, .jpg formats). For suggestions on figure preparation/layout, please refer to our Figure Preparation Guidelines: <http://bit.ly/EMBOPressFigurePreparationGuideline>
- 5) Point-by-point response letters should include the original referee comments in full together with your detailed responses to them (and to specific editor requests if applicable), and also be uploaded as editable (e.g., .docx) text files.
- 6) Please complete our Author Checklist, and make sure that information entered into the checklist is also reflected in the manuscript; the checklist will be available to readers as part of the Review Process File. A download link is found at the top of our Guide to Authors: embopress.org/page/journal/14602075/authorguide
- 7) All authors listed as (co-)corresponding need to deposit, in their respective author profiles in our submission system, a unique ORCID identifier linked to their name. Please see our Guide to Authors for detailed instructions.
- 8) Please note that supplementary information at EMBO Press has been superseded by the 'Expanded View' for inclusion of additional figures, tables, movies or datasets; with up to five EV Figures being typeset and directly accessible in the HTML version of the article. For details and guidance, please refer to: embopress.org/page/journal/14602075/authorguide#expandedview
- 9) To facilitate reproducibility and cross-laboratory adoption of methodologies, please structure the Materials & Methods section as outlined in our guide to authors, including a completed Reagents and Tools Table that can be downloaded from our author guidelines as well (<https://www.embopress.org/page/journal/14602075/authorguide#structuredmethods>).
- 10) Digital image enhancement is acceptable practice, as long as it accurately represents the original data and conforms to community standards. If a figure has been subjected to significant electronic manipulation, this must be clearly noted in the figure legend and/or the 'Materials and Methods' section. The editors reserve the right to request original versions of figures and the original images that were used to assemble the figure. Finally, we generally encourage uploading of numerical as well as gel/blot image source data; for details see: embopress.org/page/journal/14602075/authorguide#sourcedata

At EMBO Press, we ask authors to provide source data for the main manuscript figures. Our source data coordinator will contact you to discuss which figure panels we would need source data for and will also provide you with helpful tips on how to upload and organize the files.

Revision to The EMBO Journal should be submitted online within 90 days, unless an extension has been requested and approved by the editor; please click on the link below to submit the revision online before 26th May 2025:
Link Not Available

If you choose to alternatively have this study further considered by another EMBO Press publication, please use the following hyperlink to directly transfer the manuscript, optionally with inclusion of referee reports and identities:
Link Not Available

Referee #1:

Telomeres are transcribed to produce the long non-coding telomere repeat containing RNA TERRA. TERRA promoters are located within the subtelomeric region of chromosome ends and drive transcription through the telomere repeats themselves on the C-rich strand. TERRA transcripts can not only hybridize with telomeric repeats in cis, but are also believed to associate with telomeres in trans. Thus, TERRA expression has been linked to the formation of RNA:DNA hybrids, or R-loops, and have long been thought to promote replication stress at telomeric DNA. However, heterogeneity in transcription initiation, transcript length, RNA processing, and stability have made the study of TERRA function a challenge. Over the years several groups have published studies looking to manipulate TERRA expression and degradation to understand how defects in TERRA regulation

contribute to telomere dysfunction.

In this manuscript, In et al. have created a CRISPRi system in HeLa cells using gRNA that can induce expression of TERRA at a group of telomere ends with the overarching goal of analyzing how TERRA affects telomere stability. The authors demonstrate that TERRA induction drives the formation of R-loops at telomere ends and that this process is dependent on RAD51 and RAD51AP1, factors that have been previously established to regulate TERRA R-loop formation. Moreover, these TERRA R-loops induce a marker of DNA damage 53BP1 and generate a fragile telomere phenotype often associated with replication stress. The fragile telomere phenotype appears to be enriched at both leading and lagging strands and as a result bypassing of this damage may be regulated by several enzymes including PrimPol, POLD3, SMARCAL1, and MUS81. The authors conclude that TERRA R-loops induce replication fork stalls that can be repaired through downstream priming of the stalled replication fork by PrimPol or cleavage and processing to drive DNA polymerase δ -mediated break-induced replication. Overall, the manuscript highlights a new technology to induce TERRA transcription, but the major findings are not novel and at times the conclusions overstate the data as they are presented.

Major Critiques:

R-loops cause replication stress during head-on collisions between the replication machinery and the transcription bubble. Given that TERRA associates with the C-rich strand, these transcription-replication collisions are likely to be co-directional. Co-directional collisions are far less damaging, acting as temporary impediment, slowing fork progression, but not leading to overt replication stress or fork stalling. However, repeated transcription bursts at a given loci, as perhaps demonstrated here, could pose a greater challenge for the replication machinery. However, whether the model system used in these studies is physiologically representative is not clear. The authors should include an additional control in Figure 1 to highlight the accumulation of R-loops in the TERRA OE cells. Do the TERRA OE cells have similar levels of telomeric R-loops as ALT cells? Other genes that are prone to R-loop formation (i.e. RPL13A)?

There is a significant amount of 53BP1 foci in these cells, in addition to sites of DSBs, 53BP1 can represent the accumulation of under-replicated DNA in G1. Are the 53BP1 positive TIF representing DSBs? Under-replicated DNA? Are these 53BP1 positive foci also positive for H2AX? Does this damage lead to overall decrease in viability with TERRA OE?

POLD3 is an essential gene, depletion of POLD3 alone should induce defects in telomere replication (really instability throughout the genome) and ultimately lethality. Likewise, the combined depletion with PRIMPOL would also be lethal making this data underwhelming.

The authors suggest that the lagging strand damage is being caused by g-quadruplex formation on the G-rich strand opposing the R-loop, but there is no data to support this conclusion. The authors should analyze G-quadruplex formation in the TERRA OE system or use another methodology to highlight lagging strand damage beyond the FISH assay.

One of the major conclusions of the paper is that TERRA OE engage BIR at telomeres, demonstrated through telomere fragility as a surrogate. Telomere fragility is a fairly vague assay that doesn't provide much insight towards the true nature of the DNA damage and is not necessarily representative of BIR. The authors should demonstrate BIR through engagement of DNA Polymerase and EdU/Telomere FISH, BrdU pull-down followed by dot blot, TRF assay demonstrating telomere length heterogeneity, etc.

Given that most R-loops occur in a co-directional orientation wouldn't another hypothesis be that defects in replication fork progression by loss of SMARCAL1 or POLD3 inhibit the clearance of RNA-DNA hybrids by the replisome which would inherently exacerbate telomere fragility. The data suggest that TERRA R-loop formation causes replication stress, but isn't plausible that the accumulation of R-loops is a consequence of ongoing stress? Perhaps this is supported by the data presented in Figure 5 and Supplemental Figure S4. These data demonstrate that combined depletion of PrimPol and POLD3 leads to a reduction in telomere fragility, but the reduction in telomere fragility seems independent of R-loops given that there is no change in TERRA R-loops in PrimPol and POLD3 deficient cells? Nor in the SMARCAL1/ PrimPol and MUS81/ PrimPol?

The proposed model needs clarification. The right branch of the model is misleading, but perhaps labels would help clarify. The way the BIR event is drawn has a newly replicated G-rich strand, invading and hybridizing with another G-rich strand? The C-strand on the gray structures appears to then be filled in by POLD3, that scenario would be unlikely given that there is no primer on the 3' distal strand to initiation DNA synthesis by POLD3?

The data in the siNT samples for the fragile telomere experiments are identical. Separate experiments should have separate controls. The data for siNT control and siNT TERRA OE in Figure 3B, E, Figure 4E, Figure 5A, B, D, Supplemental Figure 3D are identical.

Referee #2:

In this manuscript the authors investigate TERRA and R-loops using an elegant CRISPR system of activating TERRA transcription by targeting a transcriptional activator to TERRA promoter regions. TERRA is typically repressed in S-phase, but this repression is absent in ALT cells, with R-loops promoting telomere extension by BIR. The authors identify that TERRA induction in telomerase-positive HeLa cells results in RAD51 and RAD51AP1 dependent R-loop formation that disrupts replication dynamics to promote HDR. The authors conclude that TERRA-induced HDR relies on PrimPol for repriming, and is both independent of and synthetic lethal with telomeric BIR in cells with high levels of TERRA.

1. Abstract - present tense. Change to "TERRA induces PrimPol-dependent repair".
2. The authors state that the induced 50-fold increase in TERRA in HeLa cells is comparable to ALT cells. Please either provide a reference for this or include an ALT cell line control.
3. Interestingly, RAD51 and RAD51AP1 depletion only decrease overexpressed TERRA R-loops, not endogenous TERRA R-loops. The explanation for this is that the basal levels of R-loops are potentially formed post transcriptionally in HeLa cells with relatively long telomeres (compared to HeLas with short telomeres). It seems somewhat surprising to me that there is no change at all. It is also interesting that RAD51 and RAD51AP1 depletion causes an increase in telomere fragility in cells with endogenous TERRA. This seems to highlight the complexity and fine-tuned regulation of TERRA R-loop formation.
4. Because R-loops are thought to interrupt the replication machinery, it would be useful to include telomere/pRPA foci (in addition to telomere/53BP1 foci).
5. Co-depletion of PrimPol and POLD3 suppressed TERRA induced telomere fragility, but caused an increase in baseline telomere fragility. This should be discussed. I am not clear on the reason for the differences (throughout the manuscript) between cells with baseline TERRA/R-loops and cells with induced TERRA/R-loops.
6. The data show that both PrimPol (repriming) and POLD3 (BIR) are simultaneously required for repair of TERRA-induced telomere damage. Would you not expect these pathways to be mutually exclusive?
7. "This suggests that the increased formation of G-quadruplex structures, prompted by TERRA R-loops, also necessitates the BIR-mediated conservative repair pathway". This seems a bit speculative without directly addressing G4 formation.
8. In Fig 6D, could HeLa be labelled similarly to U2OS in 6E?
9. It would be interesting to interrogate further the mechanism of cell death following PrimPol and POLD3 depletion in HeLa and U2OS cells. S4C shows there is no change in cell cycle distribution, but is this at an early timepoint? It would also be nice to see this effect more broadly in terms of correlation with TERRA levels, ie in other cell lines with variable TERRA levels or titrated overexpression of TERRA?
10. The telomere length and ALT biomarker data should be included, as this is important to the overall conclusions, and seems an obvious omission.
11. The data demonstrating BIR are not sufficiently convincing.
12. The title is misleading, as telomere maintenance has not been demonstrated.
13. The study is somewhat limited by looking at a limited number of phenotypes following specific genetic depletions, more thorough characterisation of telomere maintenance and BIR would be useful.

Referee #3:

This manuscript sheds light on new repair mechanisms to overcome telomeric R-loops-induced damage. TERRA is the long non-coding RNA transcribed at telomeres starting from the subtelomeric regions at the chromosome ends. It is known to form RNA-DNA hybrids both co-transcriptionally and in trans by strand invading the target telomeric DNA in a Rad51-dependent manner. The role of TERRA and its RNA-DNA hybrids is still elusive; however, it is known that increased RNA-DNA hybrids at telomeres causes DNA damage and homology-directed repair (HDR) which is important for telomere maintenance specifically in the alternative lengthening mechanism of telomeres (ALT).

The authors develop a novel approach to induce endogenous TERRA overexpression in HeLa cells and study the downstream effects. By using dCAS9 fused with the transcription activator VP16 along with guide RNAs targeting the subtelomeric promoters of TERRA, they were able to enhance the transcript production at several subtelomeres. As expected, TERRA RNA-DNA hybrids were also increased along with the associated DNA damage. Interestingly, the authors demonstrate the co-existence of two repair mechanisms to re-initiate DNA synthesis upon blockade at TERRA RNA-DNA hybrids sites: one that utilizes the enzyme PrimPol and semiconservative DNA replication and one that engages the conservative break-induced replication (BIR) mechanism.

The manuscript holds potential in terms of novelty, not only for the methods used but also for the experimental findings. The authors explained in a clear and concise manner the workflow deriving reasonable conclusions. The strength of the paper lies in four major points of novelty for the telomere-associated research field:

- 1) Development of a new tool to induce endogenous TERRA overexpression and accumulation of RNA-DNA hybrids;

- 2) Telomeric RNA-DNA hybrids form in trans also upon induced endogenous TERRA overexpression
- 3) TERRA may be used by PrimPol to re-initiate DNA synthesis upon fork stalling at sites of RNA-DNA hybrids;
- 4) BIR and PrimPol-mediated replication co-exist to repair RNA-DNA hybrids-induced damage.

Figure 1 the authors show the efficacy of the new TERRA-overexpressing system by measuring the levels of the transcript and RNA-

DNA hybrids globally and individually at single telomeres with and without induction. It is claimed that the levels of TERRA produced by HeLa cells upon induction are similar to the ones normally present in U2OS cells. This observation may be important to draw parallels with telomere maintenance during ALT. Therefore, it would be important to show in the supplementary figures that the levels of TERRA induced in HeLa cells truly correspond to the ones observed in U2OS cells.

Figure 3, the manuscript focuses on the DNA damage associated with TERRA overexpression highlighting that the severity of distress is similar to the one caused by zeocin treatment. In the IF-FISH pictures, telomeres foci upon TERRA overexpression seem slightly bigger than the uninduced control. Did the authors measure the telomere length of HeLa cells upon TERRA induction and compare it to the control?

Figure 4, the authors show that TERRA overexpression induces telomere fragility at both the leading and lagging strands. In the text, they explain the function of the RNase H1 enzyme which has been used already in previous figures. It would make more sense to introduce this explanation earlier in the manuscript.

Figure 6, absence of PrimPol and POLD3 leads to growth impairment of both HeLa and U2OS cells. It would be interesting to extend the time frame of the study and perhaps monitor marks of DNA damage at telomeres at multiple time points.

We would like to thank all three referees for their very careful, insightful, thoughtful and objective evaluation of our manuscript. The points are well taken and we will try to address them as well as possible upon revision.

Referee #1:

Telomeres are transcribed to produce the long non-coding telomere repeat containing RNA TERRA. TERRA promoters are located within the subtelomeric region of chromosome ends and drive transcription through the telomere repeats themselves on the C-rich strand. TERRA transcripts can not only hybridize with telomeric repeats in cis, but are also believed to associate with telomeres in trans. Thus, TERRA expression has been linked to the formation of RNA:DNA hybrids, or R-loops, and have long been thought to promote replication stress at telomeric DNA. However, heterogeneity in transcription initiation, transcript length, RNA processing, and stability have made the study of TERRA function a challenge. Over the years several groups have published studies looking to manipulate TERRA expression and degradation to understand how defects in TERRA regulation contribute to telomere dysfunction.

In this manuscript, In et al. have created a CRISPRi system in HeLa cells using gRNA that can induce expression of TERRA at a group of telomere ends with the overarching goal of analyzing how TERRA affects telomere stability. The authors demonstrate that TERRA induction drives the formation of R-loops at telomere ends and that this process is dependent on RAD51 and RAD51AP1, factors that have been previously established to regulate TERRA R-loop formation. Moreover, these TERRA R-loops induce a marker of DNA damage 53BP1 and generate a fragile telomere phenotype often associated with replication stress. The fragile telomere phenotype appears to be enriched at both leading and lagging strands and as a result bypassing of this damage may be regulated by several enzymes including PrimPol, POLD3, SMARCAL1, and MUS81. The authors conclude that TERRA R-loops induce replication fork stalls that can be repaired through downstream priming of the stalled replication fork by PrimPol or cleavage and processing to drive DNA polymerase δ -mediated break-induced replication. Overall, the manuscript highlights a new technology to induce TERRA transcription, but the major findings are not novel and at times the conclusions overstate the data as they are presented.

Response: We kindly disagree with the statement that the major findings are not novel and do our best to explain the novelty in the revised version of the manuscript. The most important novel findings are the following:

- We demonstrate for the first time that RAD51 and RAD51AP1 play non-redundant roles for post transcriptional R-loop formation. The Zou-paper (Yadav et al. 2022) proposed that R-loops are formed by RAD51AP1 independently of RAD51 while Feretzaki et al. (2020) demonstrated the requirement of RAD51. We resolve the discrepancy of the Feretzaki and Zou-papers.
- We demonstrate that TERRA R-loops induce break-induced replication (BIR) and PRIMPOL-dependent repair in NON-ALT cells. So-far the effects of TERRA R-loops were studied in the context of ALT cells in which BIR is already activated, by unknown mechanisms and which have strongly altered histone composition as well as very different telomeric protein composition at telomeres. It has never been demonstrated before that TERRA per se can induce BIR.
- We demonstrate that TERRA R-loops in S phase induce two parallel pathways of repair, BIR and PRIMPOL. We demonstrate that these pathways are critical for the survival of telomerase cells with high TERRA levels as well as an ALT cancer cell. PRIMPOL-dependent repair of at telomeres has never been demonstrated before.

Major Critiques:

R-loops cause replication stress during head-on collisions between the replication machinery and the transcription bubble. Given that TERRA associates with the C-rich strand, these transcription-replication collisions are likely to be co-directional. Co-directional collisions are far less damaging, acting as temporary impediment, slowing fork progression, but not leading to overt replication stress or fork stalling. However, repeated transcription bursts at a given loci, as perhaps demonstrated here, could pose a greater challenge for the replication machinery. However, whether the model system used in these studies is physiologically representative is not clear. The authors should include an additional control in Figure 1 to highlight the accumulation of R-loops in the TERRA OE cells. Do the TERRA OE cells have similar levels of telomeric R-loops as ALT cells? Other genes that are prone to R-loop formation (i.e. RPL13A)?

Response: As requested, we have now included in New Figure 1B data in which we quantify and compare TERRA levels in ALT cells with TERRA OE cells (quantification of TERRA signal normalized to rRNA by Dot blot; HeLa, HeLa with induced TERRA expression, U2OS, VA13, GM847, SAOS-2 ALT cells). The data confirm that the TERRA OE HeLa cells express TERRA in a similar range as ALT cancer cells.

There is a significant amount of 53BP1 foci in these cells, in addition to sites of DSBs, 53BP1 can represent the accumulation of under-replicated DNA in G1. Are the 53BP1 positive TIF representing DSBs? Under-replicated DNA? Are these 53BP1 positive foci also positive for H2AX? Does this damage lead to overall decrease in viability with TERRA OE?

Response: We demonstrate in Figure 6D that TERRA overexpression in wild type cells does not impact on cell viability. However, upon abolishment of PRIMPOL and POLD3-dependent repair, the viability is strongly reduced. Thus, cells can cope with high R-loop levels at telomeres in S phase as long as they can engage the characterized repair pathways. As requested, we have characterized the TIFs further: we have determined if ATM kinase, which is activated by double strand DNA breaks, or ATR kinase, which is activated by single stranded DNA, is responsible for the TIFs (New Figure 3D). Importantly, our data demonstrate that both, inhibition of ATR and inhibition of ATM reduce the TIFs in TERRA OE cells. Activation by ATM (by DNA double strand breaks) and ATR (by single stranded DNA) are consistent with the data in the paper and the model put forward in Figure 7C. TERRA R-loops induce first breakage of replication forks by MUS81 (ATM activation) followed by 5' strand resection (ATR activation) which precedes strand invasion for BIR. Furthermore, we demonstrate that phosphorylated RPA32, which is a hallmark of single stranded DNA damage and replication stress, accumulates at telomeres upon TERRA induction, providing additional support for the involvement of ATR.

POLD3 is an essential gene, depletion of POLD3 alone should induce defects in telomere replication (really instability throughout the genome) and ultimately lethality. Likewise, the combined depletion with PRIMPOL would also be lethal making this data underwhelming.

Response: we demonstrate that TERRA-overexpressing cells are impaired in their viability much more strongly than non-overexpressing cells. Thus, TERRA induces damage that requires these pathways for cell viability, which we consider as major findings.

The authors suggest that the lagging strand damage is being caused by g-quadruplex formation on the G-rich strand opposing the R-loop, but there is no data to support this conclusion. The authors should analyze G-quadruplex formation in the TERRA OE system or use another methodology to highlight lagging strand damage beyond the FISH assay.

Response: We cite the Zou-paper (Yadav et al. 2022) which demonstrated G-quadruplex formation upon formation of TERRA R-loops. We removed the referral to G-quadruplexes on Page 17. We refer to the potential contributions of G-quadruplexes to lagging strand telomere fragility stating it as speculation on pages 11-12 and the legend to Figure 7C. We do agree that further characterization of lagging strand fragility by TERRA would have been useful. However, because of the following reasons we did not do this. 1. The effects of G-quadruplex structures on telomere structure and maintenance have been described before in mutant cells lacking certain G-quadruplex unwinding DNA helicases. Therefore, a characterization of TERRA R-loop induced G-quadruplex structures would be interesting but less novel. 2. TERRA hybridizes with the telomeric DNA strand replicated by leading strand synthesis and our study focuses on the damage which occurs during synthesis of the leading strand (Figure 7C). The induction of G-quadruplexes by TERRA on the lagging strand telomeres is a collateral damage, which stems from the fact that the in the R-loop displaced G-rich telomeric DNA strand when not base-paired with the C-rich telomeric DNA strand can form G-quadruplexes. 3. There are practical issues: The lead author, postdoc Suna In is leaving the lab by the end of May. We tried with all efforts to carry out the experiments, which we considered to be most urgent. Furthermore, in the context of another project by another postdoc, we have been unable so-far to reproduce a published assay which detects G-quadruplexes in cells. Thus, it is not a straightforward experiment. At this point we are still struggling with the assay and I'm not sure if and when we will succeed.

One of the major conclusions of the paper is that TERRA OE engage BIR at telomeres, demonstrated through telomere fragility as a surrogate. Telomere fragility is a fairly vague assay that doesn't provide much insight towards the true nature of the DNA damage and is not necessarily representative of BIR. The authors should demonstrate BIR through engagement of DNA Polymerase η and EdU/Telomere FISH, BrdU pull-down followed by dot blot, TRF assay demonstrating telomere length heterogeneity, etc.

Response: From the work published by Sullivan and co-workers (<http://dx.doi.org/10.1016/j.celrep.2016.10.048>) we understand that DNA Polymerase η depletion enhances ALT activity rather than reducing it. Thus, DNA Polymerase η , while playing a role in the regulation of replication stress and ALT, appears not to be a necessary co-factor for BIR. However, as suggested we have carried out EdU/Telomere FISH experiments, to test telomere synthesis in G2 cells representing a hallmark of BIR. The new data (New Figures 6D, E, F) demonstrate that TERRA OE induces EdU-positive telomeres in G2 as expected for BIR. We now also emphasize more clearly and explain better on page 16-17 the importance of our CO-FISH results (Figure 6C), which demonstrate conservative DNA synthesis of telomeric DNA

in TERRA overexpressing cells. Importantly, conservative DNA synthesis in which both DNA strands are newly synthesized is another hallmark of BIR. Both of these data provide very strong support for engagement of BIR. They re-enforce our results obtained upon POLD3, SMARCAL1 and MUS81 depletion.

Given that most R-loops occur in a co-directional orientation wouldn't another hypothesis be that defects in replication fork progression by loss of SMARCAL1 or POLD3 inhibit the clearance of RNA-DNA hybrids by the replisome which would inherently exacerbate telomere fragility. The data suggest that TERRA R-loop formation causes replication stress, but isn't plausible that the accumulation of R-loops is a consequence of ongoing stress? Perhaps this is supported by the data presented in Figure 5 and Supplemental Figure S4. These data demonstrate that combined depletion of PrimPol and POLD3 leads to a reduction in telomere fragility, but the reduction in telomere fragility seems independent of R-loops given that there is no change in TERRA R-loops in PrimPol and POLD3 deficient cells? Nor in the SMARCAL1/ PrimPol and MUS81/ PrimPol?

Response: We demonstrate that TERRA R-loops induce telomere fragility. We demonstrate that TERRA R-loops are not influenced by SMARCAL1 and PRIMPOL. Furthermore, we find that SMARCAL1 and POLD3 depletion reduces telomere fragility – it does NOT enhance it as stated by the referee! Thus, SMARCAL1 and POLD3 function downstream of TERRA R-loops. We now emphasize this point more clearly on pages 14 and 15, SMARCAL1 and POLD3-dependent repair induces telomere fragility.

It is interesting to note that TERRA R-loops pose an obstacle for the replisome despite the fact that they are formed on the strand that is bound by the CMG replicative DNA helicase. Though this helicase is able to unwind R-loops at other regions in the genome, it appears to fail at telomeres when encountering TERRA R-loops. This conclusion is supported by this as well as previous papers published by several laboratories. TERRA R-loop induced fragility does also not depend on collisions between RNA and DNA polymerases. Indeed, we demonstrated that TERRA expressed from plasmids forms post-transcriptional R-loops at telomeres in trans, which induce telomere fragility (Feretzi et al. 2020). R-loops formed post transcription in trans cannot lead to transcription-replication collisions.

The proposed model needs clarification. The right branch of the model is misleading, but perhaps labels would help clarify. The way the BIR event is drawn has a newly replicated G-rich strand, invading and hybridizing with another G-rich strand? The C-strand on the gray structures appears to then be filled in by POLD3, that scenario would be unlikely given that there is no primer on the 3' distal strand to initiate DNA synthesis by POLD3?

Response: We very thankful to the referee for pointing out the errors in the model and apologize for our oversight. We have now redrawn Figure 7 clarifying our results and conclusions.

The data in the siNT samples for the fragile telomere experiments are identical. Separate experiments should have separate controls. The data for siNT control and siNT TERRA OE in Figure 3B, E, Figure 4E, Figure 5A, B, D, Supplemental Figure 3D are identical.

Response: All these experiments were done together and not separately. However, we depicted the results in separate Figures in order to explain the different phenotypes. We have clarified this point in the revised Figure legends. We thank the referee for pointing this out.

Referee #2:

In this manuscript the authors investigate TERRA and R-loops using an elegant CRISPR system of activating TERRA transcription by targeting a transcriptional activator to TERRA promoter regions. TERRA is typically repressed in S-phase, but this repression is absent in ALT cells, with R-loops promoting telomere extension by BIR. The authors identify that TERRA induction in telomerase-positive HeLa cells results in RAD51 and RAD51AP1 dependent R-loop formation that disrupts replication dynamics to promote HDR. The authors conclude that TERRA-induced HDR relies on PrimPol for repriming, and is both independent of and synthetic lethal with telomeric BIR in cells with high levels of TERRA.

1. Abstract - present tense. Change to "TERRA induces PrimPol-dependent repair".

Response: We have corrected the tense, thank you!

2. The authors state that the induced 50-fold increase in TERRA in HeLa cells is comparable to ALT cells. Please either provide a reference for this or include an ALT cell line control.

Response: As requested, we have now included in New Figure 1B data in which we quantify and compare TERRA levels in ALT cells with TERRA OE cells (quantification of TERRA signal normalized to rRNA by Dot blot; HeLa, HeLa

with induced TERRA expression, U2OS, VA13, GM847, SAOS-2 ALT cells). The data confirm that the TERRA OE HeLa cells express TERRA in a similar range as ALT cancer cells.

3. Interestingly, RAD51 and RAD51AP1 depletion only decrease overexpressed TERRA R-loops, not endogenous TERRA R-loops. The explanation for this is that the basal levels of R-loops are potentially formed post transcriptionally in HeLa cells with relatively long telomeres (compared to HeLas with short telomeres). It seems somewhat surprising to me that there is no change at all. It is also interesting that RAD51 and RAD51AP1 depletion causes an increase in telomere fragility in cells with endogenous TERRA. This seems to highlight the complexity and fine-tuned regulation of TERRA R-loop formation.

Response: We agree with these statements. We tried our best to discuss them in the text indicating the involvement of repair factors in potentially dealing with fragility stemming from G-quadruplex structures, unresolved t-loops, oxidative and other DNA damage,

4. Because R-loops are thought to interrupt the replication machinery, it would be useful to include telomere/pRPA foci (in addition to telomere/53BP1 foci).

Response: We have characterized the TIFs further: we have now determined that ATM kinase, which is activated by double strand DNA breaks, and ATR kinase, which is activated by single stranded DNA and replication stress, are responsible for the TIFs (New Figure 3D). Specifically, our new data demonstrate that both, inhibition of ATR and inhibition of ATM reduce the TIFs in TERRA OE cells. Activation by ATM (by DNA double strand breaks) and ATR (by single stranded DNA) are consistent with the data in the paper and the model put forward in Figure 7C. TERRA R-loops induce first breakage of replication forks by MUS81 (ATM activation) followed by 5' strand resection (ATR activation) which precedes strand invasion for BIR. We have also determined that phosphorylated RPA accumulates at telomeres, which is indicative of replication stress-induced ATR-signaling.

5. Co-depletion of PrimPol and POLD3 suppressed TERRA induced telomere fragility, but caused an increase in baseline telomere fragility. This should be discussed. I am not clear on the reason for the differences (throughout the manuscript) between cells with baseline TERRA/R-loops and cells with induced TERRA/R-loops.

Response: We tried to explain this in the text and tried to improve the wording in the revised version. Basal levels of telomere fragility may stem from several replication stress sources including G-quadruplex structures, unresolved t-loops and oxidative and other DNA damage. Resolving these problems requires a very large number of factors which have only partially been characterized. Upon TERRA induction, we strongly increase telomere fragility which allowed us to identify the factors and mechanisms that repair TERRA R-loop-induced damage. Some of the characterized factors may also be involved in overcoming TERRA-independent telomere replication obstacles which cannot be distinguished in the control cells with low TERRA levels.

6. The data show that both PrimPol (repriming) and POLD3 (BIR) are simultaneously required for repair of TERRA-induced telomere damage. Would you not expect these pathways to be mutually exclusive?

Response: The two pathways work in parallel. Both can engage to overcome the TERRA R-loops induced damage. The CO-FISH analysis indicates that ~5% of leading strand telomeres are repaired by conservative DNA synthesis (Fig. 6C; see leading strand telomeric loss) which is consistent with BIR. On the other hand, the CO-FISH analysis of telomere fragility indicated that due to overexpressed TERRA ~7.5% of leading strand telomeres are repaired by semiconservative DNA replication (see Fig. 4H, subtract fragility in siNT in control to siNT in TERRA OE cells), which is consistent with PRIMPOL-dependent repair. We therefore estimate that both pathways are similarly contributing to overcoming TERRA-induced replication stress. We now explain this observation on page 16-17.

7. "This suggests that the increased formation of G-quadruplex structures, prompted by TERRA R-loops, also necessitates the BIR-mediated conservative repair pathway". This seems a bit speculative without directly addressing G4 formation.

Response: We cite the Zou-paper (Yadav et al. 2022) which demonstrated G-quadruplex formation upon formation of TERRA R-loops and we change the wording: we replace 'suggest' by 'speculate' in the relevant sentence. We do agree that further characterization of lagging strand fragility by TERRA would have been useful. However, because of the following reasons we did not do this. 1. The effects of G-quadruplex structures on telomere structure and maintenance have been described before in mutant cells lacking certain G-quadruplex unwinding DNA helicases. Therefore, a characterization of TERRA R-loop induced G-quadruplex structures would be interesting but less novel. 2. TERRA hybridizes with the telomeric DNA strand replicated by leading strand synthesis and our study focuses on the damage which occurs during synthesis of the leading strand (Figure 7C). The induction of G-quadruplexes by TERRA on the lagging strand telomeres is a collateral damage, which stems from the fact that the in the R-loop

displaced G-rich telomeric DNA strand when not base-paired with the C-rich telomeric DNA strand can form G-quadruplexes. 3. There are practical issues: The lead author postdoc, Suna In is leaving the lab by the end of May. We tried with all efforts to carry out the experiments, which we considered to be most urgent. Furthermore, in the context of another project by another postdoc, we have been unable so-far to reproduce a published assay which detects G-quadruplexes in cells. Thus, it is not a straightforward experiment. At this point we are still struggling with the assay and I'm not sure if and when we will succeed.

8. In Fig 6D, could HeLa be labelled similarly to U2OS in 6E?

Response: We have added the label to now Fig. 7A, thank you!

9. It would be interesting to interrogate further the mechanism of cell death following PrimPol and POLD3 depletion in HeLa and U2OS cells. S4C shows there is no change in cell cycle distribution, but is this at an early timepoint? It would also be nice to see this effect more broadly in terms of correlation with TERRA levels, ie in other cell lines with variable TERRA levels or titrated overexpression of TERRA?

Response: We now clarify in the Figure legend that the cell cycle distribution was determined at day 4 which corresponds to the latest time point. Titration would be interesting but tricky in our experimental system.

10. The telomere length and ALT biomarker data should be included, as this is important to the overall conclusions, and seems an obvious omission.

Response: We have now added the requested data in New Figures EV5C-F.

11. The data demonstrating BIR are not sufficiently convincing.

Response: We have now carried out EdU/Telomere FISH experiments, to test telomere synthesis in G2 cells representing a hallmark of BIR. The new data (New Figures 6D, E, F) demonstrate that TERRA OE induces EdU-positive telomeres in G2 as expected for BIR. We now also emphasize more clearly and explain better the importance of our CO-FISH results (Figure 6C), which demonstrate conservative DNA synthesis of telomeric DNA in TERRA overexpressing cells. Importantly, conservative DNA synthesis in which both DNA strands are newly synthesized is another hallmark of BIR. Both of these data provide very strong support for engagement of BIR. They re-enforce our results obtained upon POLD3, SMARCAL1 and MUS81 depletion.

12. The title is misleading, as telomere maintenance has not been demonstrated.

Response: We show in the CO-FISH experiments of Figure 6C that TERRA induces conservative replication of telomeres. i.e. both DNA strands are newly synthesized. This is consistent with telomere breakage and de novo telomere synthesis by BIR and contrasts the canonical semiconservative DNA replication of telomeres. We have emphasized this point more clearly in the revised version of the paper on pages 16-17.

13. The study is somewhat limited by looking at a limited number of phenotypes following specific genetic depletions, more thorough characterisation of telomere maintenance and BIR would be useful.

Response: We demonstrate involvement of POLD3, MUS81 and SMARCAL1. As indicated in the response to point 11, we have also carried out EdU/Telomere FISH experiments. We agree it will be interesting to test additional factors that have been implicated in BIR but estimate that such studies go beyond the scope of the current manuscript.

Referee #3:

This manuscript sheds light on new repair mechanisms to overcome telomeric R-loops-induced damage. TERRA is the long non-coding RNA transcribed at telomeres starting from the subtelomeric regions at the chromosome ends. It is known to form RNA-DNA hybrids both co-transcriptionally and in trans by strand invading the target telomeric DNA in a Rad51-dependent manner. The role of TERRA and its RNA-DNA hybrids is still elusive; however, it is known that increased RNA-DNA hybrids at telomeres causes DNA damage and homology-directed repair (HDR) which is important for telomere maintenance specifically in the alternative lengthening mechanism of telomeres (ALT).

The authors develop a novel approach to induce endogenous TERRA overexpression in HeLa cells and study the downstream effects. By using dCAS9 fused with the transcription activator VP16 along with guide RNAs targeting the

subtelomeric promoters of TERRA, they were able to enhance the transcript production at several subtelomeres. As expected, TERRA RNA-DNA hybrids were also increased along with the associated DNA damage. Interestingly, the authors demonstrate the co-existence of two repair mechanisms to re-initiate DNA synthesis upon blockade at TERRA RNA-DNA hybrids sites: one that utilizes the enzyme PrimPol and semiconservative DNA replication and one that engages the conservative break-induced replication (BIR) mechanism.

The manuscript holds potential in terms of novelty, not only for the methods used but also for the experimental findings. The authors explained in a clear and concise manner the workflow deriving reasonable conclusions. The strength of the paper lies in four major points of novelty for the telomere-associated research field:

- 1) Development of a new tool to induce endogenous TERRA overexpression and accumulation of RNA-DNA hybrids;
- 2) Telomeric RNA-DNA hybrids form in trans also upon induced endogenous TERRA overexpression
- 3) TERRA may be used by PrimPol to re-initiate DNA synthesis upon fork stalling at sites of RNA-DNA hybrids;
- 4) BIR and PrimPol-mediated replication co-exist to repair RNA-DNA hybrids-induced damage.

Figure 1 the authors show the efficacy of the new TERRA-overexpressing system by measuring the levels of the transcript and RNA-DNA hybrids globally and individually at single telomeres with and without induction. It is claimed that the levels of TERRA produced by HeLa cells upon induction are similar to the ones normally present in U2OS cells. This observation may be important to draw parallels with telomere maintenance during ALT. Therefore, it would be important to show in the supplementary figures that the levels of TERRA induced in HeLa cells truly correspond to the ones observed in U2OS cells.

Response: As requested, we have now included in New Figure 1B data in which we quantify and compare TERRA levels in ALT cells with TERRA OE cells (quantification of TERRA signal normalized to rRNA by Dot blot; HeLa, HeLa with induced TERRA expression, U2OS, VA13, GM847, SAOS-2 ALT cells). The data confirm that the TERRA OE HeLa cells express TERRA in a similar range as ALT cancer cells. TERRA OE HeLa cells have similar TERRA levels as SAOS-2 ALT cells. They are slightly lower than in U2OS cells and slightly higher than in VA13 and GM847 ALT cells.

Figure 3, the manuscript focuses on the DNA damage associated with TERRA overexpression highlighting that the severity of distress is similar to the one caused by zeocin treatment. In the IF-FISH pictures, telomeres foci upon TERRA overexpression seem slightly bigger than the uninduced control. Did the authors measure the telomere length of HeLa cells upon TERRA induction and compare it to the control?

Response: We measured telomere length on Southern blots which did not change in a detectable manner (New Figure EV5C).

Figure 4, the authors show that TERRA overexpression induces telomere fragility at both the leading and lagging strands. In the text, they explain the function of the RNase H1 enzyme which has been used already in previous figures. It would make more sense to introduce this explanation earlier in the manuscript.

Response: We have changed the text as suggested, thank you.

Figure 6, absence of PrimPol and POLD3 leads to growth impairment of both HeLa and U2OS cells. It would be interesting to extend the time frame of the study and perhaps monitor marks of DNA damage at telomeres at multiple time points.

Response: This experiment would be interesting but would require a new setup. We transiently deplete PRIMPOL and POLD3 with siRNAs and even by this treatment most TERRA-overexpressing cells die by day 4.

Dr. Joachim Lingner
EPFL
ISREC
Station 19
Lausanne 1015
Switzerland

23rd Jun 2025

Re: EMBOJ-2025-120125R
TERRA R-loops trigger a switch in telomere maintenance towards break-induced replication and PRIMPOL-dependent repair

Dear Joachim,

Thank you for submitting your final revised manuscript for our consideration. I am pleased to inform you that in light of the positive re-reviews (below), we have now accepted it for publication in The EMBO Journal.

With kind regards,

Hartmut

Referee #2:

The authors have satisfactorily addressed all my comments. I feel the manuscript content and the clarity of the findings are substantially improved. I support publication.

Referee #3:

The authors have addressed all major concerns. This report uses a new system of TERRA induction to clarify existing literature findings and extend them substantially by implicating PRIMPOL and BIR in resolving R-loop encounters.